# Genetic subtypes of smoldering multiple myeloma are associated with distinct pathogenic phenotypes and clinical outcomes

Mark Bustoros [1,2,15], Shankara Anand[3,4,15], Romanos Sklavenitis-Pistofidis[1,3], Robert Redd [5],
Eileen M. Boyle [6], Benny Zhitomirsky[3], Andrew J. Dunford[3], Yu-Tzu Tai [1], Selina J. Chavda[7], Cody Boehner[1],
Carl Jannes Neuse[1,8], Mahshid Rahmat [1], Ankit Dutta [1], Tineke Casneuf[9], Raluca Verona[10],
Efstathis Kastritis[11], Lorenzo Trippa[5], Chip Stewart[3], Brian A. Walker [12], Faith E. Davies[6],
Meletios-Athanasios Dimopoulos [11], P. Leif Bergsagel [13], Kwee Yong [7], Gareth J. Morgan [6],
François Aguet [3], Gad Getz [3,14,15✉] & Irene M. Ghobrial [1,3,15✉]

Smoldering multiple myeloma (SMM) is a precursor condition of multiple myeloma (MM) with significant heterogeneity in disease progression. Existing clinical models of progression risk do not fully capture this heterogeneity. Here we integrate 42 genetic alterations from 214 SMM patients using unsupervised binary matrix factorization (BMF) clustering and identify six distinct genetic subtypes. These subtypes are differentially associated with established MM-related RNA signatures, oncogenic and immune transcriptional profiles, and evolving clinical biomarkers. Three genetic subtypes are associated with increased risk of progression to active MM in both the primary and validation cohorts, indicating they can be used to better predict high and low-risk patients within the currently used clinical risk stratification models.

[1] Medical Oncology, Dana-Farber Cancer Center, Boston, MA, USA. [2] Division of Hematology & Medical Oncology, Meyer Cancer Center, Weill Cornell Medicine, New York, NY, USA. [3] Broad Institute of MIT & Harvard, Cambridge, MA, USA. [4] Boston University School of Medicine, Boston, MA, USA. [5] Biostatistics and Computational Biology, Dana-Farber Cancer Institute, Boston, MA, USA. [6] Perlmutter Cancer Center, NYU Langone Health, New York, NY, USA. [7] Division of Hematology, University College London, London, UK. [8] University of Münster Medical School, Münster, Germany. [9] Janssen Research and Development, Beerse, Belgium. [10] Janssen Research and Development, Spring House, PA, USA. [11] Department of Clinical Therapeutics, National and Kapodistrian University of Athens, Athens, Greece. [12] Melvin and Bren Simon Comprehensive Cancer Center, Indiana University, Indianapolis, IN, USA. [13] Division of Hematology, Mayo Clinic, Scottsdale, AZ, USA. [14] Department of Pathology, Massachusetts General Hospital Cancer Center, Boston, MA, USA. [15] These authors contributed equally: Mark Bustoros, Shankara Anand, Gad Getz, Irene M. Ghobrial. ✉email: gadgetz@broadinstitute.org; Irene_Ghobrial@dfci.harvard.edu

Multiple Myeloma (MM) is an incurable plasma cell malignancy with significant inter- and intra-patient heterogeneity. It is almost always preceded by the asymptomatic precursor stages monoclonal gammopathy of undetermined significance (MGUS) and smoldering multiple myeloma (SMM). Approximately 1.5% of MGUS patients will progress to MM per year, while SMM patients have a higher overall progression risk of 10% per year[1,2]. Like MM, SMM is a heterogeneous condition—some patients have over a 50% risk of progression within two years, while others have more MGUS-like disease that grows slowly[3].

Several risk stratification models exist to help clinicians differentiate patients with high risk of progression to active myeloma from those for whom a "watchful waiting" approach is appropriate. The existing models rely solely on clinical measurements, many of which are indicators of tumor burden and universal biomarkers of MM for risk stratification. These models, however, do not fully partition progressors from non-progressors, and patients classified as low- or intermediate-risk still progress to active MM and have a 2-year progression risk of 6% and 18%, respectively (compared to 44% for high-risk patients)[3], which warrants more accurate models that also represent the molecular heterogeneity in MM. We recently showed that genomic alterations in mitogen-activated protein kinase (MAPK) and DNA repair pathways or *MYC* are independently predictive of progression risk[4]. While these genomic biomarkers improved upon the clinical models, they represent only a few alterations that do not capture the full extent of genetic heterogeneity in SMM.

Multiple myeloma is characterized by multiple chromosomal gains or losses, structural variations, driver single nucleotide variations (SNVs)[5–7], and other structural alterations involving known oncogenes[8]. The IgH translocations and copy number alterations (CNAs) are considered early events in the pathogenesis of MM, while other CNAs and SNVs usually occur later during clonal evolution, providing more proliferative capacity to the tumor cells[4]. Multiple SMM studies have shown that CNAs, including whole chromosome duplications and arm-level losses or gains, are the most common events, followed by SNVs, and then translocations[4,7,9,10]. These alterations have all been detected at the SMM stage, and we previously showed that the genomic makeup of the MM tumor clone is fully acquired by the time of SMM diagnosis in most cases[4,10–12]. Furthermore, certain genetic alterations were found to occur more frequently together[4,7], therefore, studying these genetic alterations as groups or networks rather than individual risk factors may improve our understanding of disease molecular groups and the overall risk stratification in SMM.

In this study, we apply an unsupervised binary matrix factorization (BMF) clustering method to identify groups of genomic alterations that tend to occur together, and show that the resulting clusters represent distinct biological and clinical subtypes in SMM.

## Results

### Identification of six clusters with distinct genetic features.
We leveraged DNA sequencing data from a cohort of 214 patients at the time of SMM diagnosis, with matched RNA sequencing data of 89 patients from the same cohort, and baseline clinical information for the whole cohort (Fig. 1A and Supplementary Table 1A, B). The patients in this cohort harbored a median of seven driver events, several of which tended to co-occur[4,6,7], suggesting that additional analyses may reveal distinct patterns (i.e., clusters) of genetic alterations. To identify these patterns, binarized DNA features (42 driver SNVs, CNVs, and translocations) were curated for each sample representing the presence or

absence of each genomic alteration. Previous work successfully subtyped diffuse large B-cell lymphoma patients with consensus non-negative matrix (NMF) factorization of numeric DNA genetic features[13]. We instead apply consensus BMF to accommodate these binarized DNA features (Supplementary Fig. 1 A), appropriately model summative features that span multiple subtypes (i.e., hyperdiploidy), and handle sparse matrices[14]. We identified six SMM patient subtypes with distinct patterns of drivers. Tumor samples in four of these clusters were hyperdiploid (more than 48 chromosomes), while those in the other two were enriched for known MM IgH translocations (Fig. 1B, Supplementary Fig. 1B, C).

Cluster 1: the tumors of this cluster exhibited a hyperdiploid genotype as the primary event and were significantly enriched in *NRAS, TRAF3,* and *MAX* mutations. We named this cluster Hyperdiploid-like 1 (HL1). Cluster 2: the tumors of this cluster harbored frequent arm-level deletions, including 16q, 6q, 1p, 17p, 4q, 18q, and 20q, and the IgH translocation t(14;20), which upregulates the transcription factor *MAFB*. Moreover, mutations in *NRAS, BRAF, TP53, ATM, MAFB,* and *CDKN2C* genes were enriched in this subgroup. Hyperdiploidy was detected in 69% of the tumors in this cluster. We named this cluster Hyperdiploid-like 2 (HL2). The tumors of this cluster were significantly enriched in deletion(16q), which involves *CYLD* tumor suppressor and other genes. The presence of both hyperdiploidy and t(14;20) in the same cluster could be explained by the co-occurrence of those events as described in prior studies[7,10]. Indeed, half of patients with t(14;20) also had hyperdiploidy in their tumor samples. Cluster 3: Tumors of this cluster exhibited primary events such as t(4;14), which upregulates *FGFR3* and *MMSET* genes; t(14;16), which upregulates the transcription factor *MAF*; and copy number losses of 14q, 1p, 8p, 10p, 11q, 12p, and 17p. We named this cluster Translocation-like 1 (TL1). This cluster was also enriched for hypodiploid tumors, defined as having fewer than 45 chromosomes (adjusted $P = 0.04$). Tumors in this cluster also harbored mutations in *DIS3, MAF, FGFR3, PRKD2, PRDM1,* and *HIST1H1E*. Many of these proteins and mutations in their encoding genes are essential to tumor cell survival and play roles in protein translation, secretion, and plasma cell differentiation[7,9,15]. Differential gene expression analysis revealed that *TL1* tumors have downregulation of ribosomal proteins and the negative regulator of the MAPK pathway *TRAF2*. The upregulated genes included *WHSC1(MMSET), FGFR3, KLHL4, CCND3,* and genes involved in the endoplasmic reticulum (ER) stress response (Fig. 2A). Cluster 4: this cluster comprises tumors with a hyperdiploid genotype that harbored mutations in KRAS and NFKBlA genes, and MYC translocations as the only significant features. We named this cluster *Hyperdiploid-like 3* (HL3). Cluster 5: the tumors in this cluster mainly exhibited t(11;14), *CCND1* mutations, and gain of chromosome 11 or its long arm. We named this cluster Translocation-like 2 (TL2). Interestingly, this cluster had significantly lower M-protein levels and was enriched in light-chain disease compared to the other clusters ($P < 0.001$ for both). Tumors of TL2 had 243 differentially expressed genes ($q < 0.1$, $\log_2 FC | > 1.5$; 180 upregulated, 63 downregulated), including overexpression of *CCND1, ERBB4, E2F7, E2F1, TRAK2, RBL1,* and downregulation of *DUSP4, TRAF6, PRKD3, CCDC6,* and *ZNF844*. Furthermore, this cluster had the highest expression of *CCND1* compared to the other clusters (Fig. 2B). Cluster 6: this is a hyperdiploid cluster similar to *HL1*; however, its tumors are also enriched in *NFKB2* and *KLHL6* mutations and exhibit copy gains in 2p. Interestingly, copy gains of 1q were more frequent in this cluster than *HL1* and the other hyperdiploid clusters ($P < 0.001$ for both comparisons). We named this cluster *Hyperdiploid-like 4* (HL4). Additionally, key individual genes in

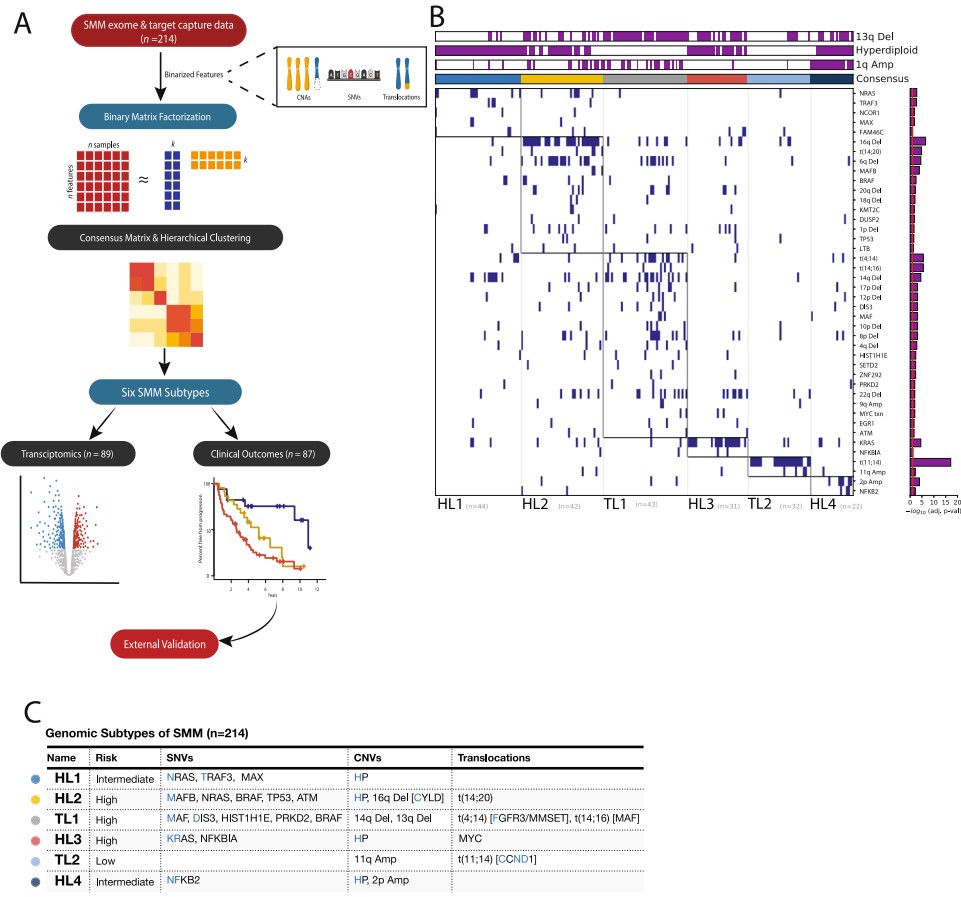

**Fig. 1 Outline of the study and the six molecular subtypes identified based on DNA alterations in tumor samples from smoldering myeloma patients.**
**A** Flowchart of analyses was performed in this study. Clusters were generated based on the tumor genetic alterations from DNA sequencing data, then they were analyzed for correlations with transcriptomic and clinical data. This flowchart was created with BioRender.com. **B** Identification of groups of tumors with corresponding genetic events. Binary matrix factorization consensus clustering was performed using somatic mutations, somatic copy number alterations, and translocations from 214 SMM tumor samples (columns). Clusters HL1–4, TL1, and TL2 with their associated landmark genetic alterations are visualized (boxed for each cluster). Genetic alterations that were positively associated with each cluster were identified by a one-sided Fisher test and ranked by significance (Benjamini–Hochberg adjusted *p* value < 0.1, red line, bar graph to the right). **C** Summary table of the six subtypes identified with selected enriched genetic features.

myeloma pathogenesis were overexpressed in tumors of specific genetic subtypes. *MCL1* was upregulated in all the genetic subgroups with the lowest expression observed in *HL1* tumors compared to the other subtypes ($P = 0.001$) (Fig. 2C). *MYC* oncogene was also highly expressed in the four hyperdiploid clusters ($P = 0.009$, Wilcoxon Test) (Fig. 2D). Cyclin D1 (*CCND1*) was significantly upregulated in *TL2* tumors ($P = 0.0001$), while *CCND2* was upregulated in *TL1* and *HL2* tumors compared to the rest of the genetic subtypes ($P = 0.004$) (Fig. 2D, Supplementary Fig. 2A, B). Moreover, in the four hyperdiploid clusters, we found that *CCND2* expression was higher in samples without 11q gain, while *CCND1* expression was higher in tumors with 11q gain (Supplementary Fig. 2D, G).

**The genetic subgroups are enriched with specific MM expression signatures**. To date, ten distinct RNA expression signatures have been defined and validated as prognostic in newly diagnosed and relapsed MM patients[16,17]. Each expression signature was then associated with specific primary genetic lesions identified by fluorescent in situ hybridization (FISH), including hyperdiploidy and IgH translocations that activate c-MAF and *MAFB*, *CCND1*, *CCND3*, or *MMSET*[16,17]. We asked whether these expression signatures were present in our SMM cohort and correlated with the six genetic subgroups. To address this, we performed a gene-

set enrichment analysis of these expression signatures among the genetic subtypes (lower panel of Fig. 2G). We observed that the hyperdiploid expression signature[16,17], which is seen in hyperdiploid MM patients, is upregulated in the tumors of our hyperdiploid clusters (HL1–4). The Cyclin D (CD) expression signatures, including CD-1 that highly expresses *CCND1* and CD-2, which expresses the B cell markers *CD20*, *CD79A*, and *CCND1* were significantly upregulated in the *TL2* genetic subgroup. Moreover, the high-risk MMSET (MS) molecular signature, which is enriched in patients with *t(4;14)* and upregulates *MMSET* and *FGFR3* genes, was upregulated in the *TL1* cluster. The MAF (MF) signature, which has been reported in patients with *t(14;16)* and *t(14;20)* that upregulate *MAF* and *MAFB* genes, respectively, was enriched in both the *TL1* and *HL2* subgroups, consistent with the presence of these genetic alterations in their tumors. The low bone disease signature, which has not been previously mapped to a specific MM genetic alteration, was upregulated in the *HL4*, *TL1*, and *HL2* subgroups, suggesting it could be linked to 1q gain, which occurs frequently in these three subgroups. Interestingly, the PR signature, which is found in proliferative tumor cells, was enriched in the *HL3* and *TL2* subgroups. Furthermore, the NFkB signature was upregulated only in *HL2*, which could be explained by the high frequency of 16q deletions and *CYLD* mutations in this subgroup. Finally, the

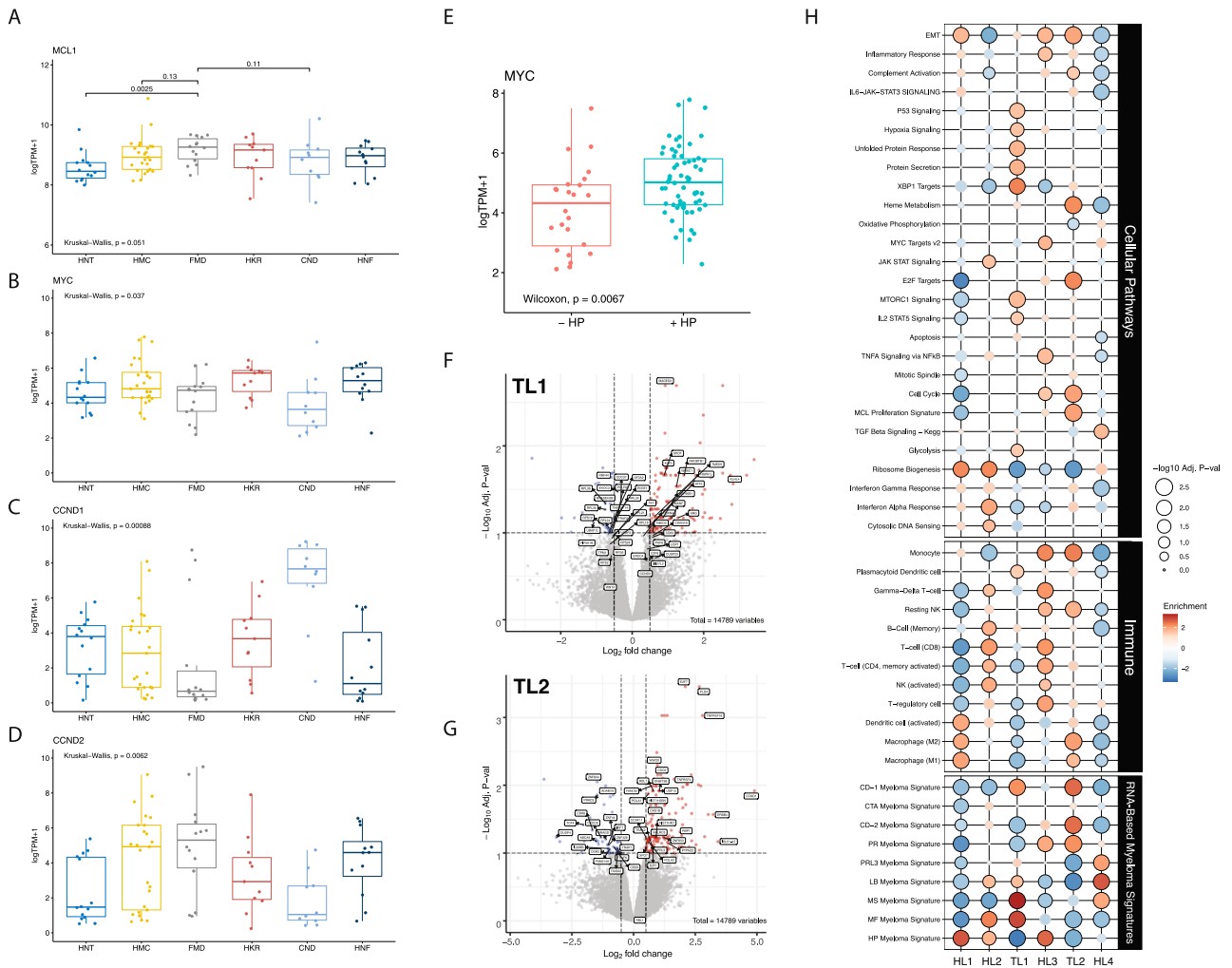

**Fig. 2 Differential gene expression and gene set enrichment analysis for tumors from the six genetic subtypes ($n = 89$). A–D** Comparison of gene expression levels of *MCL1, MYC, CCND1,* and *CCND2* among the six genetic subtypes. Two-sided $p$ value derived from Kruskal–Wallis test. **E** Comparison of expression levels of *MYC* oncogene expression between the two non hyperdiploid ones (-HL) and the four hyperdiploid (+HL) subtypes. Two-sided $p$-value was calculated using the Wilcoxon rank-sum test. Expression is measured by the $log_2$ value of transcript per million of each gene ($log_2 TPM + 1$). Boxplots representing median, and interquartile range, whiskers representing first, and fourth quartile. **F** Volcano plot showing fold changes for genes differentially expressed between tumors of TL1 subtype and the other subtypes. **G** Volcano plot showing fold changes for genes differentially expressed between tumors of TL2 subtype and the other subtypes. *X* axis = $Log_2$ fold change, *Y* axis = $-log_{10}$ adjusted $p$ value. **H** Gene set enrichment analysis of different molecular and oncogenic pathways (top), immune cell signatures (middle), and MM-specific signatures (bottom) among the six genetic subtypes.

PRL3 signature, which overexpresses the protein tyrosine phosphatase *PTP4A3* and 27 additional genes, was upregulated only in *HL4* and indicates that it could also be linked to the presence of 1q gain, which is found in all the tumors of the *HL4* subgroup.

We further examined whether our genetic subtypes were enriched in specific mutational signatures for 72 samples with matched normal whole-exome sequencing. We found that the APOBEC mutational signature activity (SBS 2,13 COSMIC v3.0) differed between the genetic subtypes ($P = 0.027$, Kruskal-Wallis) while AID mutational signatures did not ($P = 0.17$) (Supplementary Fig. 2E–G). Specifically, we found APOBEC activity enriched in the *HL2* & *TL1* clusters vs. the rest of tumors ($P = 0.006$, *adjusted p = 0.045*) (Supplementary Fig. 2H).

**Genetic subgroups have distinct transcriptional profiles**. We performed GSEA on the available transcriptomic resulting data to explore which genes and biological pathways were differentially expressed among the genetic subgroups we identified. Pathways that were significantly enriched within the six genetic subtypes

are described and illustrated (Fig. 2G). We found that protein secretion, unfolded protein response (UPR), glycolysis, hypoxia, and mTOR signatures were specifically enriched in the *TL1* subgroup, while E2F target genes, cell cycle, heme metabolism, complement, and proliferation signatures were enriched in *TL2* tumors. Genes induced by MYC were highly expressed in *HL3* and *HL4*, consistent with MYC upregulation in these two clusters. The NFkB, cytosolic DNA sensing, and JAK-STAT signatures were enriched in the tumors of *HL2*. The interferon-alpha and gamma response signatures were high in *HL2* but low in *TL1*. Interestingly, oxidative phosphorylation, WNT-beta-catenin, and TGF-beta signaling were enriched only in tumors of *HL4*, and the TNFa and inflammatory signatures were uniquely enriched in *HL3*. The ribosome biosynthesis signature was low in *TL1, TL2,* and *HL3* but high in *HL1, HL2,* and *HL4* subgroups.

We also looked at signatures related to the tumor immune microenvironment. Signatures of regulatory T cells and NK cells were high in *HL2* and *HL3*, while the M2 macrophage signature was high in *TL2* and *HL4* tumors. The *HL3* and *TL2* tumors were enriched for the monocyte signature. In contrast, the signature of

plasmacytoid dendritic cells, known for their immunosuppressive effect[18], was only enriched in the *TL1* tumors.

**Genetic subtypes are differentially associated with risk of progression and evolving clinical biomarkers**. To investigate the relationship between these genetic subtypes and clinical outcome, we analyzed a subset of patients ($n = 87$) who were followed for the natural course of their disease and did not receive any treatment in a clinical trial setting before progression to MM. Their baseline characteristics are reported in Supplementary Table 2. The median follow-up time for these patients was 7.1 years and the median time to progression (TTP) was 4 years (95% CI, 3–6). In this cohort, 57 patients (66%) have progressed, while 30 (34%) remained asymptomatic as of the last follow-up (put date of last follow-up in the methods section). The genetic subgroups had different outcomes, measured by TTP to active MM (log-rank $P = 0.007$) (Supplementary Fig. 3A). Median TTP for patients in *HL2*, *TL1*, and *HL3* was 3.7, 2.6, and 2.2 years, respectively, while it was 4.3, 11, and 5.2 years for *HL1*, *TL2*, and *HL4*, respectively. The *HL2*, *TL1*, and *HL3* genetic subgroups had increased hazards of progression (HR > 4.5) to active myeloma (Supplementary Fig. 3B).

We then divided the genetic subtypes based on their TTP and hazards of progression into high- (*HL2*, *TL1*, *HL3*), intermediate- (*HL1*, *HL4*), and low-risk (*TL2*) subtypes. The high- and intermediate-risk subtypes had significantly shorter TTP and increased risk of progression compared to the low-risk subtype (2.6 and 5.2 vs. 11 years, respectively, $P < 0.0001$) (Fig. 3A). We also stratified the patients according to the 20-2-20 model, which uses three cutoffs of M-protein >2 g/dL, FLC ratio >20, and bone marrow plasmacytosis >20% to define low, intermediate, and high-risk groups based on the presence of none, one, and two or all these variables, respectively[3]. The intermediate- and high-risk genetic subtypes and the clinically high-risk SMM group (according to the 20-2-20 model) were the only significant predictors of progression to active MM in our multivariate analysis (Fig. 3B). Moreover, the prediction performance of the combined clinical and genetic models was higher than the clinical model alone (C-index: 0.76 vs 0.71, respectively). Interestingly, within the high-risk clinical group, patients in the high-risk genetic subgroups had increased progression risk (HR 3.7, 95% CI:1.1–12.8, $P = 0.04$). We observed a similar finding in the intermediate-risk clinical group, where patients from the high-risk genetic groups had shorter TTP compared to intermediate and low-risk ones (3.4 vs. 6.5, and 10.9 years, respectively, $P = 0.003$, with a two-year progression risk of 33% vs. 0%, respectively). (Supplementary Fig. 3C). We observed that high-risk genetic subtypes were significantly enriched with specific genetic alterations, such as *KRAS*, *TP53*, *t(4;14)*, *t(14;16)*, 1q gain, 16q, and 1p deletions among others (Supplementary Table 3).

We also identified patients with evolving M protein (eMP), which is defined as a ≥ 25% increase in M-protein within 12 months of diagnosis with a minimum absolute increase of 0.5 g/dL, and evolving hemoglobin (eHb), which is defined as a ≥ 0.5 g/dl decrease within 12 months of diagnosis[19]. These changing patterns were reported to confer a higher risk of progression to active MM in different SMM cohorts. We found that the odds of eMP and eHb were 9.4 and 5.3 times higher ($P = 0.006$ and 0.007, respectively) in patients with the high-risk genetic subtypes. These results indicate that high-risk genetic subgroups have distinct genetic and transcriptomic features as well as different clinical outcomes in terms of progression to active MM and evolution of its biomarkers over time.

**Validation of the molecular subtypes in external cohorts**. To validate our findings on the clinical significance of the genetic subtypes, we developed a classifier based on the features of the clusters we identified in our primary cohort (Supplementary Fig. 4). We used an external cohort of 75 SMM patients to validate the classifier and investigate whether the genetic subtypes are predictive for progression[11]. The patients in this cohort were enriched in the low-risk clinical stage and had a median TTP of 5 years. Similar to the primary cohort, patients in the intermediate and high-risk genetic subtypes had increased risk of progression to active MM in multivariate analysis accounting for the clinical risk stage (HR: 4.5 and 9, $P = 0.039$ and 0.002, respectively) (Fig. 3C). We found that adding the genetic risk groups improved the prediction of progression compared to the clinical model only (C-index: 0.76 vs 0.65, respectively) (Supplementary Table 4). We also obtained another smaller cohort of 67 patients with targeted capture data, including common MM translocations, CNAs and SNVs, and added it to the previous cohort[12]. In those 142 patients, *HL2*, *TL1*, *HL3*, and *HL4* subtypes were independent predictors of progression to active myeloma (Fig. 3D) and the high-risk genetic subtypes were associated with increased risk of progression in multivariate analysis (HR: 3.4, 95% CI: 1.68–6.7). We then asked, given the small number of patients in the different cohorts, whether combining the three cohorts would provide more power and increase the significance of our genetic classification. The combined cohort contained 229 SMM patients with median follow-up and TTP of 6.9 and 5.2 years, respectively. Indeed, the genetic subtypes had a different TTP (Fig. 3F), and the high-risk genetic subtypes had significantly shorter TTP compared to the low or the intermediate-risk groups (Fig. 3F). We also found that both the individual genetic subtypes and the genetic risk groups were independent predictors of progression in the combined cohort multivariate analysis, validating our initial findings (Fig. 3G). Interestingly, within the high-risk clinical stage, patients in the low-risk genetic subgroups had significantly lower progression risk (HR 0.26, 95% CI: 0.1–0.6, $P = 0.001$) and median TTP of 8.7 years (log-rank $P = 0.002$) (Supplementary Fig. 5A). In the intermediate-risk clinical group, patients from the high-risk genetic groups had increased risk of progression to symptomatic MM (HR 4.4, 95% CI: 1.7–11.6, $P = 0.002$) and shorter TTP (3 vs 6.9 and 9.4 years, respectively, log-rank $P = 0.001$) (Supplementary Fig. 5B).

## Discussion

This study modeled the genetic heterogeneity seen in SMM by identifying genetic subtypes that correspond to phenotypic attributes and clinical outcomes, providing a deeper understanding of SMM pathogenesis. We and others have previously cataloged individual driver genetic aberrations in SMM and MM cohorts[4,11,12]. However, the present study expands on this work and identifies SMM genetic subtypes defined by multiple recurrent DNA genetic aberrations, unlike previous classification efforts that were mainly based on gene expression data. Our findings suggest that these genetic subtypes could have distinct evolutionary histories depending on the initiating genetic events (translocations or CNAs), which may influence the subsequent acquisition of cooperating genetic aberrations.

The defined genetic subtypes had distinct clinical outcomes of disease progression into symptomatic MM, which could provide us with comprehensive molecular models for predicting progression and dynamic changes in clinical biomarkers over time. They also have specific dysregulated molecular and oncogenic pathways, which could facilitate the identification of specific targets and selection of therapies for each genetic subtype to

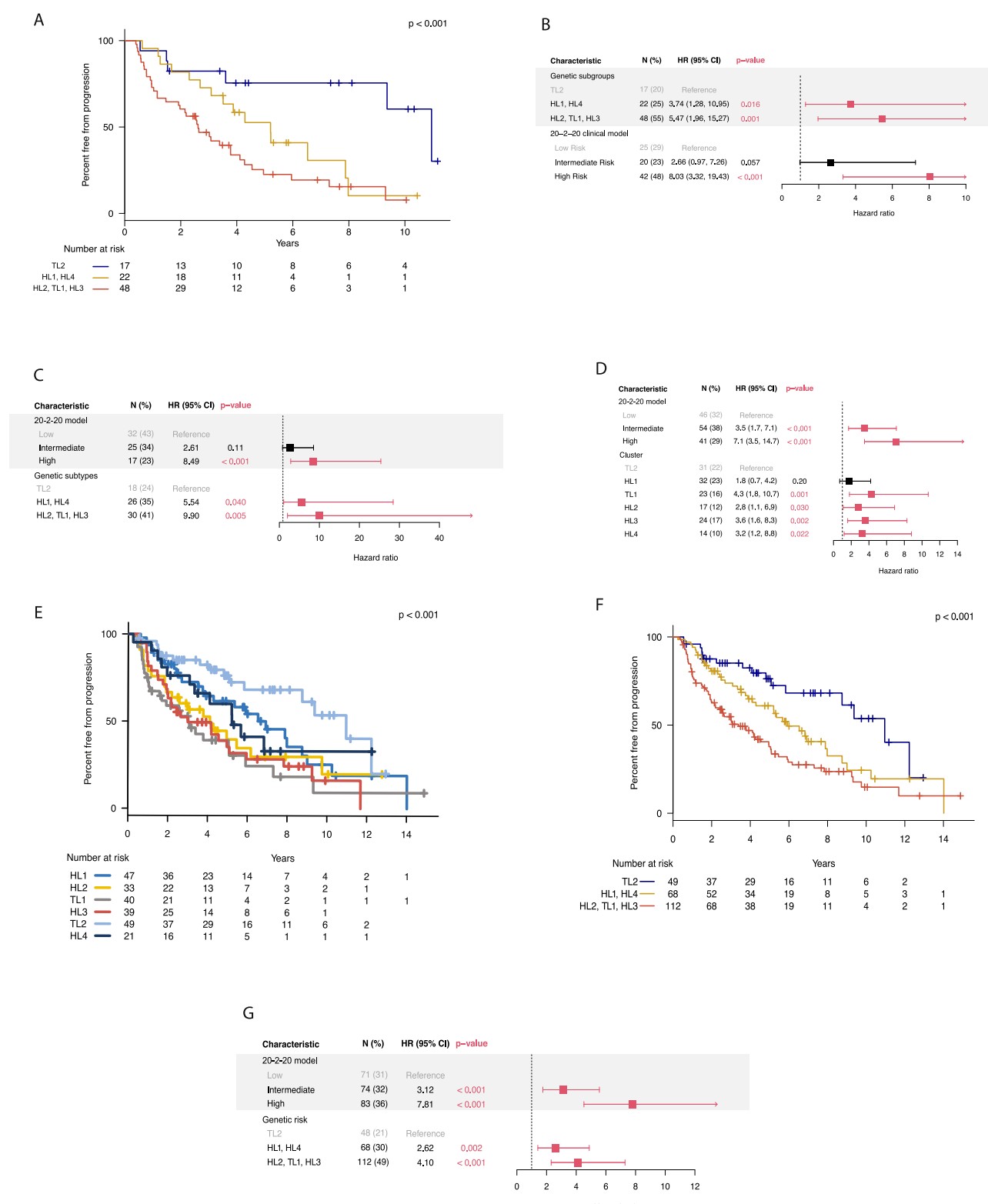

empower precision medicine efforts, much like the specific efficacy of venetoclax in patients with *t*(11;14)[20,21].

We identified six clusters based on the detected genetic alterations. We divided them into three high-risk (*HL2, TL1, HL3*), two intermediate-risk (*HL1, HL4*), and one low-risk (*TL2*) genetic group based on progression risk to active MM. We found that DNA repair aberrations were exclusive to *HL2* and *TL1* subgroups, which were enriched in *TP53* mutations and deletions.

Also, *MYC* expression was higher in the hyperdiploid subgroups than the non-hyperdiploid ones, consistent with previous reports of a higher frequency of *MYC* alterations in hyperdiploid MM patients[22]. The key Cyclin D genes, *CCND1* and *CCND2*, were highly expressed in *TL2* and *TL1*, respectively. *CCND1* and *CCND2* expression patterns were previously reported to distinguish between MM patients hyperdiploid tumor samples[23]; indeed, in the four hyperdiploid clusters, we found the former to

**Fig. 3 Clinical outcomes of the six molecular and the risk groups in the primary and validation cohorts. A** Kaplan-Meier curves for analysis of TTP in patients ($n = 87$) belonging to the three genetic risk groups (Low: TL2, Intermediate: HL1, HL4, High: HL2, TL1, HL3); log-rank $p$ value = 0.0005. **B** Multivariate cox regression analysis of the low, intermediate, and high-risk genetic subtypes and clinical risk stages according to the IMWG 20/2/20 model in the primary cohort ($n = 87$). **C** Multivariate cox regression analysis of the low, intermediate, and high-risk genetic subtypes and clinical risk stages according to the IMWG 20/2/20 model in the first validation cohort ($n = 74$). **D** Multivariate cox regression analysis of the genetic subtypes and clinical risk stages according to the IMWG 20/2/20 model in the two validation cohorts ($n = 142$). **E** Kaplan-Meier curves for analysis of TTP in patients from the six genetic subtypes in the combined cohort ($n = 229$); log-rank $p$ value = 0.002. **F** Kaplan-Meier curves for analysis of TTP in patients belonging to the three genetic risk groups of the combined cohort ($n = 229$); log-rank $p$ value = 0.0002. **G** Multivariate cox regression analysis of the low, intermediate, and high-risk genetic subtypes and clinical risk stages according to the IMWG 20/2/20 model in the combined cohorts ($n = 229$). Forest plots are used to visualize the multivariate analysis. IMWG International Myeloma Working Group, $N$ number of patients with event and percentages from the total number of patients evaluable, HR hazards ratio, error bars indicate 95% CI. All $p$ values are two-sided. Differences in survival curves and subsequent two-sided $p$ values were calculated using the log-rank test.

be enriched in tumors with 11q gain, while the latter is highly expressed in tumors without 11q gain. However, we could not assess their prognostic impact due to the small number of samples with gene expression data in patients who were followed for their disease course.

The gene expression signatures of specific molecular and oncogenic processes also varied significantly between the genetic subgroups. For example, *TL1* tumors showed specific enrichment for protein secretion, ER stress, UPR, glycolysis, and mTOR signaling. This molecular phenotype manifested clinically where patients with this genetic subtype had the highest increase in M-protein levels at six and twelve months from diagnosis. Such patients may benefit from drugs inducing cellular stress, such as proteasome inhibitors or novel molecules targeting the ER stress and UPR pathways[24]. Alternatively, *TL2* tumors were uniquely enriched with genes related to B-lymphocytes, cell cycle, heme metabolism, and complement activation signaling. Clinically, these patients had the longest TTP, lowest baseline M-protein level, and the least increase over time. We also found that the *HL2* tumors were enriched for interferon-alpha response, cytosolic DNA sensing, and JAK-STAT signatures. These results underscore the phenotypic difference among the genetic subtypes and provide a conceptual framework for future functional studies that aim to validate or therapeutically target the dysregulated pathways and tumor dependencies in different genetic subtypes.

In our multicenter cohort, we found that the three high-risk subgroups (*HL2*, *TL1*, and *HL3*) had an increased risk of progression and were associated with evolving hemoglobin and M-protein levels, showing that these subgroups are also predictive of the dynamic changes in MM clinical biomarkers over time. The high-risk genetic subtypes were independent risk factors of progression to overt MM after accounting for the clinical risk stage by the 20-2-20 model. Moreover, among those patients considered high- and intermediate-risk by this model, those with the high-risk genetic subtypes progressed faster to active myeloma than the rest in the same clinical risk group. Finally, to validate and test the significance and application of the genetic subtypes in unseen tumor samples, we trained a classifier and tested it on two external SMM cohorts and found the genetic subtypes and risk groups to be predictive for progression in those external cohorts similar to the primary cohort. Furthermore, to increase the power of our analysis, we combined the three cohorts together and found the same effect with more significance levels compared to our initial findings. Of note, the genetic features enriched in the high-risk genetic group were also found to confer a higher-risk of progression as individual features, with exception of *t*(14;16) and *t*(14;20) (Supplementary Tables 5 and 6). In fact, we and others haven't found them to confer a high risk of progression on their own[4,10,11]. However, multiple studies have shown that *t*(14;16) is frequently associated with APOBEC signature and genomic instability[4–10]. In our study, it was found in 5% of patients and

with similar rates in the validation cohorts, so larger studies with patients enriched for *t*(14;16) may be needed to confidently determine its prognostic significance in SMM. One of the limitations of our study is that we could not assess the prognostic impact of the MM signatures in comparison to our DNA subgroups because of the small number of cases with this data. Another limitation is that we depended on FISH studies in assessing MYC translocations in the primary cohort. FISH studies are less sensitive in detecting MYC translocations compared to novel targeted sequencing panels. Indeed, the validation cohorts, which used a targeted NGS panel in detecting MYC alterations, had more events compared to ours, suggesting that further studies that detect MYC with next-generation sequencing panels in SMM are needed to delineate the characteristics of tumors harboring this important feature. However, MYC alterations was a prognostic factor for progression in the primary and second validation cohorts, as well as the three combined cohorts together. Finally, we propose this genetic classification to be applied only in the SMM stage as we haven't tested its prognostic significance in active or relapsed MM settings.

In conclusion, these findings move us closer to identifying the SMM patients who are truly at a high risk of disease progression through better predictive models that integrate the molecular makeup of the tumor cells and may also guide precision medicine efforts to match targeted therapies with the optimal patient groups in multiple myeloma and its asymptomatic stages.

## Methods

**Patient samples**. We used next-generation sequencing technologies to study 214 patients with SMM at the time of diagnosis. We performed whole exome sequencing (WES) of 72 matched tumor-normal samples (mean target coverage 109×), WES on 94 tumor-only samples (with mean coverage 174×), and targeted deep sequencing on 48 samples (mean target coverage 774×). FISH data were used to determine the presence of IgH translocations. Samples were collected at Dana-Farber Cancer Institute, University College London, and the University of Athens in Greece, in addition to diagnostic samples from patients participating in phase II clinical trial for treating patients with SMM (*NCT02316106*). Patients who presented with MM symptoms at diagnosis, including hypercalcemia, renal impairment, anemia, or bone lytic lesions (CRAB), or had any myeloma-defining event were excluded from the analysis[25]. All samples were obtained after approval of the study protocols by the institutional review boards and ethics committees of the participating institutions including Dana-Farber Cancer Institute, University College London, and the University of Athens in Greece, and participating institutions of the above-mentioned clinical trial, and written informed consent from patients. All relevant ethical regulations were followed, and all the research was conducted in accordance with the Declaration of Helsinki.

**Whole exome sequencing**. Tumor DNA was extracted from CD138 + cells from patients' bone marrow samples. For germline control (normal), DNA was obtained from buccal mucosa (saliva), or peripheral blood mononuclear cells. Genomic DNA was extracted using QIAamp DNA mini kit (QIAGEN) according to the manufacturer's protocols, and double-stranded DNA concentration was quantified using PicoGreen dsDNA Assay kit (Life Technologies). Libraries were prepared by Agilent SureSelect XT2 Target Enrichment kit. To capture the coding regions, we used the SureSelect XT2 V5 + UTR capture probes (Agilent). All sequencing was performed on the Illumina HiSeq 4000 platform at the Broad Institute. For tumor

only samples ($n = 94$), libraries were prepared and hybridized using Agilent SureSelect XT2 V5 capture probes (Agilent) and sequenced on Illumina HiSeq 2500 platform.

**Targeted deep sequencing**. Genomic DNA was extracted using QIAamp DNA micro kit (QIAGEN) according to the manufacturer's protocols. The libraries for targeted sequencing were prepared using SureSelect XT Reagent Kits (Agilent), and an in-house bait set targeting 117 genes, including pan-cancer driver genes and frequently mutated genes in MM. The libraries were quantified using Agilent Tapestation, then pooled and loaded onto the Illumina HiSeq 4000 sequencer.

**Computational analysis**. The output from Illumina software was processed by the Picard data processing pipeline to yield BAM files containing well-calibrated, aligned reads. We have utilized the Broad Institute and the Getz Lab CGA WES Characterization pipeline (https://getzlab.slack.com/archives/DHC9613KQ/p1647451761073289) to call, filter, and annotate somatic mutations and copy number variation. The pipeline employs the following tools: MuTect[26], ContEst[27], Strelka[28], Orientation Bias Filter[29], DeTiN[30], AllelicCapSeg[31], MAFPoNFilter[32], ABSOLUTE[33], GATK[34], PicardTools[35], Variant Effect Predictor[36], Oncotator[37]. Recurrent sCNAs were identified using the GISTIC2.0[38]. We applied ABSOLUTE to estimate sample purity, ploidy, and absolute somatic copy numbers. These were used to infer the cancer cell fraction (CCF) of point mutations from the WES data, following the framework previously described[33]. Mutations were thereafter classified as clonal based on the posterior probability that the CCF exceeded 0.90 and subclonal if otherwise.

**Germline filtering of tumor-only cohort**. For each SNP or indel that passed all standard filters, its CCF, purity, ploidy, and local copy number were used to determine the log ratio of the probability that its allele fraction is consistent with the allele fraction modeled for a hypothetical germline event and the probability it is consistent with a modeled somatic event, as previously described[13]. After applying the Germline Somatic Log odds filter, we used the ExAC database to further exclude potential germline events that have occurred in five or more participants of any ethnic background.

**Artifact filtering**. Both cohorts were subjected to standard artifact filtering through the Broad Institute's CGA pipeline, including a TCGA panel of normal samples (PoN) filter for common germline mutations and artifacts and filters for OxoG and FFPE damage[29].

**Paired tumor-normal cohort**. We applied ABSOLUTE to estimate sample purity, ploidy, and absolute somatic copy numbers. These were used to infer the CCFs of point mutations from the WES data. We excluded 13 samples from this group due to low tumor fraction (>20%) and inconclusive FISH results. Bleed-through error associated with sequencing was observed and cleaned using a custom PoN run through the same sequencing pipeline, as described previously[32]. Two more artifacts were identified in this cohort, primarily characterized by C>A and C>T substitutions, respectively, henceforth referred to as A1 and A2. Artifact A1 was shown to represent reference bias, with a preponderance of C>A over G>T substitutions, related to oxidative damage occurring during DNA library preparation, as previously described [4]. To address this, we developed a tool that removes C>A SNPs with a low number of reads supporting the alternate allele from a sample, until the p-value of a binomial test assuming a probability of 0.5 exceeds 0.1. Of unidentified origin, artifact A2 was characterized by a preponderance of C>T SNPs in the GCC trinucleotide context over COSMIC signature 5 and was addressed by removing C>T SNPs in the GCC context with low alternate allele counts until they matched the number of C>T SNPs in the CCG context, assuming reference COSMIC signature 5 as a null. Of note, these artifacts did not affect any of the SNPs reported in genes that are frequently mutated in MM.

**Tumor-only cohort**. In this cohort, we observed an artifact of unidentified origin that was primarily characterized by T > G substitutions (Supplementary figure, panel C) and hotspot mutations in genes not reported before in multiple myeloma, henceforth referred to as A3. We addressed it by removing all hotspot mutations with less than 2 occurrences in COSMIC and do not affect genes reported to be recurrently mutated in multiple myeloma. Of note, this artifact did not affect any of the SNPs reported infrequently mutated genes in MM.

**Bulk RNA-sequencing**. Out of the 214 unique patient tumor DNA samples, there were 89 matching tumor samples material for RNA sequencing. These samples were isolated using Qiagen RNA kit. Libraries were prepared using Illumina Total mRNA kit and submitted for sequencing on Hiseq 2500 machines. We computationally processed these RNA samples using the GTEx V8 pipeline and aligned them to Hg19 Gencode v19[39]. Data quality control metrics are provided in Supplementary Fig. 6.

**Clustering approach**. A vast number of approaches have been applied to clustering multi-omic sequencing data. Late integration algorithms that cluster data types separately and then integrate them to a final result, such as COCA and PINS were previously described[40–42]. Dimensionality reduction algorithms such as jointNMF and multiNMF similarly factorize each data type separately before final integration[43,44]. iCluster is a probabilistic approach that computes a low dimensionality composition of each data modality designed to fit Gaussian distributed data[45]. However, our translocation measurement includes six sparse events, rendering any approaches requiring clustering of this data type separately or with Gaussian data assumptions difficult to apply. Furthermore, our choice of using binarized features (SNVs, CNVs, and translocations) allows us to use a simpler, "early integration" approach, where the feature space is combined before the algorithm is applied. Future work using patient similarity-based approaches, such as similarity network fusion or Cancer Integration via Multi Kernel Learning are promising future directions[46,47]. Formulating these to account for Bernoulli distributed translocation data would be ideal for including these important drivers of MM pathogenesis. As more SMM patient cohorts are gathered and sequenced with transcriptomic, proteomic, and chromatin accessibility data, applying patient similarity approaches and algorithms with more appropriate distributional assumptions is key. However, our BMF approach is appropriate for sparse, non-negative, binarized data curated in this initial cohort.

**BMF clustering workflow**. To identify patients with shared, co-occurring DNA features, we applied a variant of non-negative matrix consensus clustering algorithm adapted for binarized input and output features, Binary Matrix Factorization (BMF). Our input matrix for subtyping consisted of a combined binarized input matrix of 42 driver genes, CNVs, and 5 translocations. To select the number of clusters ($K$) for the consensus clustering, we randomly down-sampled our input matrix and computed silhouette scores using Dice dissimilarity, residuals of factorization fit, variance explained, and K-L divergence on binary matrix factorizations over a range of $K$. We found a decrease in K-L divergence with our full dataset from $K = 5$ to $K = 6$, which suggested that 6 clusters were best suited to ensure a converged factorization for $N = 214$ (Supplementary Fig. 7A–E). Additionally, we found that variance stabilized when we performed down sampling analyses at $N = 75–100$, suggesting we were powered to perform binary matrix factorization for a cohort at this minimum size. We concluded that a minimum of 100 samples and 6 clusters were suited for this approach. We take the following steps for subtyping:

1. Run BMF for our primary cohort ($n = 214$) from $K = 2$ to $K = 10$
2. Run hierarchical clustering of the consensus matrix with Euclidean distance and Ward linkage
3. Select $K = 6$ clusters from downsampling results.

We assessed binary feature importance by performing a Fisher's exact test to count feature representation within each cluster and outside of this cluster, testing for an equal proportion. Genetic alterations that were positively associated with each cluster were identified by a one-sided Fisher test and ranked by significance of Benjamini-Hochberg adjusted p value. The false discovery rate (FDR) was calculated using the Benjamini–Hochberg procedure[48].

**RNA differential expression and pathway enrichment analysis**. We performed one vs. rest gene differential expression for each identified DNA-based subtype. The limma-voom pipeline was used[49]; FDR was performed using the Benjamini-Hochberg procedure. Using genes at an FDR < 0.1, we performed ranked gene-set enrichment analysis (GSEA) using the fGSEA R package, using a rank of $-\log10(FDR) *signed-log$ Fold-Change[50]. We computed pathway enrichments for the HALLMARK and KEGG gene sets from MsigDB[51,52]. We also computed pathway enrichments for previously curated gene sets related to lymphoma and multiple myeloma[16,17,53].

**Mutational signatures**. We used the default settings of the SignatureAnalyzer tool (https://getzlab.slack.com/archives/DHC9613KQ/p1647454800384449) to extract de novo mutational signatures from a 96 base-pair context for 72 tumor-normal samples with WES[54]. Extracted signatures were mapped to Cosmic 3.0 using cosine similarity[54–56].

**Subtype Classifier**. We trained a random forest classifier on 36 overlapping translocations, SNVs, and CNVs between both our primary cohort and validation cohort found in at least 3 or more patients to predict molecular subtypes for each patient. We used scikit-learn Random Forest Classifier class[57]. The classifier reported a mean fivefold cross-validation accuracy on our primary cohort of 86.7% (SD ± 5%) after performing a randomized grid search to hypertune parameters (Supplementary Fig. 4A–C). The classifier was then used on unseen data of 142 SMM samples from two validation cohorts.

**Validation cohorts**. We obtained genomic data from two previously published studies in SMM and filtered the patients according to the updated IMWG criteria for diagnosis of patients with MM as what was done in the primary cohorts[11,12,25]. First cohort contained 75 SMM patients[11]. The second cohort contained 67 SMM

patients[12]. We applied the classifier to these cohorts to identify the genetic subtypes that each patient sample belongs to and perform subsequent analysis.

**Statistical analysis**. Binary outcomes were reported as proportions with 95% exact binomial confidence intervals. Continuous measures were summarized as median and range. Binary outcomes and other categorical variables were tested for association with continuous and other categorical variables using Wilcoxon rank-sum (or Kruskal-Wallis for three or more groups) or Fisher's exact tests, respectively. Time-to-event endpoints are estimated using the method of Kaplan and Meier, with 95% confidence intervals calculated using Greenwood's method of variance estimation. Differences in survival curves were assessed using log-rank tests. Median follow-up was calculated using the reverse Kaplan-Meier method. Unadjusted and adjusted Cox modeling was performed to assess the impact of the presence of a MM driver on clinical outcome measures, alone and in the presence of clinical features known to impact outcome. Time to progression (TTP) was measured from date of diagnosis to date of documented progression to MM. Clinical and laboratory parameters and genetic features were reported with hazard ratios and 95% confidence intervals with Wald $p$ values, while genetic features were assessed for importance of association with TTP. For comparison of clinical model only vs the clinical and genetic models, we used analysis of variance test on the Cox model with the selected variables. A global assessment of each model was also assessed using a C-statistic for censored survival data[58]. The statistic for each time-to-event model is reported with a 95% confidence interval. Values range between 0.5 to 1 indicating a poor to perfect model; nested models may be compared via overlap in the point estimates and confidence intervals. Genetic alterations that were positively associated with each cluster were identified by a one-sided Fisher test and ranked by significance of Benjamini–Hochberg adjusted p value. All other $P$ values in the study were two-sided, and adjustment for multiple hypothesis testing was performed using the method of Benjamini and Hochberg; $P$ and $q$ value thresholds for significance were set at 0.05 and 0.1, respectively. Statistical analyses were performed using R version 3.6.0 (2019-04-26).

**Reporting summary**. Further information on research design is available in the Nature Research Reporting Summary linked to this article.

## Data availability
The Genomic and transcriptomic data of the primary cohort generated in this study including the whole exome, targeted capture, and RNA sequencing data) have been deposited in the dbGAP database under accession number phs001323.v3.p1. Access to the raw data can be obtained upon request. The other published data used as validations cohorts in this study are already deposited in public databases. For the first validation cohort[11], the targeted panel data are deposited in the European Genome-phenome Archive (EGA) database under accession code EGAD00001005056. The whole-exome sequencing is deposited in the EGA database under accession code EGAD00001005285. These data are available under restricted access; access can be obtained upon request. The raw data of the published second validation cohort is deposited in the NCBI Sequence Read Archive (SRA) BioProject under accession number PRJNA541307[12]. The remaining data are available within the Article or Supplementary Information file.

## Code availability
The code for the BMF consensus clustering and the subsequent analysis in the primary and validation cohorts is available through GitHub at https://github.com/getzlab/SMM_clustering_2020[59].

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

## Acknowledgements

We would like to acknowledge Anna Justis, Ph.D. for scientific writing support and Jean-Baptiste Alberge for critical review of the manuscript. We thank all the patients and their families for their contributions to this study. This study was supported in part by grants from the National Institutes of Health (NIH; R01CA205954), Multiple Myeloma Research Foundation (MMRF) Prevention Project-Perelman Family Foundation Early Disease Translational Research Program, Leukemia and Lymphoma Society (LLS) Specialized Center of Research (SCOR) program, Dr. Miriam and Sheldon G. Adelson Medical Research Foundation, and Cancer Research UK (CRUK) Early Detection and Diagnosis Program (Grant Number: A28770). K.Y. receives funding from the National Institute for Health Research-University College London Hospitals (NIHR UCLH) Biomedical Research Centre. Selina Chavda receives funding from the Medical Research Council (MRC), UK. This research was also supported by a Stand Up To Cancer (SU2C) Dream Team grant (SU2C-AACR-DT-28-18). Stand Up To Cancer is a program of the Entertainment Industry Foundation. Research grants are administered by the American Association for Cancer Research, the scientific partner of Stand Up To Cancer. Opinions, interpretations, conclusions, and recommendations are those of the author(s) and are not necessarily endorsed by Stand Up To Cancer, the Entertainment Industry Foundation, or the American Association for Cancer Research.

## Author contributions

M.B., S.A., F.A., I.M.G., and G.G. conceived the study. M.B., S.A., F.A., R.S.P., R.R., E.K., J.P., C.J.N., S.J.C., T.C., Y.T., R.V., and P.L.B. collected the data. M.B., K.S., T.H.M., M.R., C.J.N., A.D., and C.B. performed the experiments and prepared DNA libraries. M.B., S.A., F.A., R.S.P., R.R., B.Z., A.J.D., L.T., C.S., K.Y., G.J.M., E.B., B.A.W., P.L.B., F.E.D., T.K., M.A.D., I.M.G., and G.G. analyzed the data. M.B., S.A., F.A., R.S.P., R.R., G.G., and I.M.G. wrote the manuscript. All authors contributed to the scientific discussion, reviewed, and edited the manuscript, and agreed on its content. G.G. and I.M.G. contributed equally as senior authors.

## Competing interests

There was no commercial funding for this study. M.B. has consulting roles with Takeda and Epizyme and has received honoraria from Takeda, Janssen, and Bristol Myers Squibb (BMS). E.K. has received honoraria from Amgen, Janssen, Takeda, Genesis Pharma, GSK, Pfizer, and research support from Amgen, Janssen, and Pfizer. R.J.S. is on the Data and Safety Monitoring Board of Juno and Celgene; has consulting roles with Gilead, Merck, and Astellas; and is on the Board of Directors of Kiadis. M.A.D. has received honoraria from Amgen, Celgene, Janssen, and Takeda. T.C. is an employee of Janssen R&D and holds stock in Johnson & Johnson. G.J.M. reports consulting roles with BMS, Sanofi, Karyopharm, Janssen, Roche, and Genentech. F.D. is on advisory boards for Amgen, BMS, Celgene, GSK, Janssen, Oncopeptides, Sanofi, and Takeda. P.L.B. reports consulting roles with Novartis, Amgen, Pfizer, BMS. B.W. reports research funding from BMS and Genentech, and Honoraria from Sanofi. I.M.G. has a consulting or advisory role with AbbVie, Adaptive, Amgen, Aptitude Health, Bristol Myers Squibb, GlaxoSmithKline, Huron Consulting, Janssen, Menarini Silicon Biosystems, Oncopeptides, Pfizer, Sanofi, Sognef, Takeda, The Binding Site, and Window Therapeutics; has received speaker fees from Vor Biopharma and Veeva Systems, Inc.; and her spouse is the CMO and equity holder of Disc Medicine. G.G. is a founder, consultant, and holds privately held equity in Scorpion Therapeutics, received research funding from IBM and Pharmacyclics, and is an inventor on patent applications related to ABSOLUTE, MSMuTect, MSMutSig (Title: COMPOSITIONS AND METHODS FOR CLASSIFYING TUMORS WITH MICROSATELLITE INSTABILITY, serial number: 16/640,349), MSIdetect (Title: COMPOSITIONS AND METHODS FOR TUMOR CHARACTERIZATION, serial number: PCT/US2021/058241), POLYSOLVER (Title: Polymorphic Gene Typing and Somatic Change Detection Using Sequencing Data, serial number: 15/037,394). F.A. and G.G. are inventors of a patent application for scaling computational genomics using graphics processing units (Title: Methods of Scaling Computational Genomics with Specialized Architectures for Highly Parallelized Computations and Uses Thereof, serial number: 17/284,708).
