## [Peer Review File · Nature Communications]

Genetic Subtypes of Smoldering Multiple Myeloma are associated with Distinct Pathogenic Phenotypes and Clinical OutcomesReviewers' Comments:

Reviewer #1:

Remarks to the Author:

Here Bustoros et al have performed an analysis on a set of 214 SMM patients to identify genetic subtypes that are associated with risk of progression to MM. This is a good paper that provides some information on the risk of a SMM patient on likelihood of progression to MM based on their genomics. Some minor comments are below.

Comments:

1. I find the clustering subtypes a little confusing. It is known that the primary events in myeloma (and its precursors) are the immunoglobulin translocations and hyperdiploidy. These primary events are then followed by the addition of copy number changes and mutations. Therefore, I find it surprising that some of the subgroups identified mix together the hyperdiploid and translocation groups – such as the HMC group that contains hyperdiploidy and t(14;20) patients. This doesn't make any sense. It may be that the computational approach puts them together, but etiologically these groups are very different and have separate origins. Previous papers doing similar clustering in the MM stage, using genomics or gene expression, generally always keep the translocation and hyperdiploid groups apart for this very reason. The authors should use some reasoning when deciding the groups and not just follow the informatics.
2. Previous groups looking at MM subgroups have also shown the importance of 11q+ in defining 2 subgroups of hyperdiploidy, which are associated with over-expression of CCND1, and define the CCND1/CCND2 expression groups in hyperdiploid MM. The authors do not use this important marker in their classifier. Given the importance of the gain of 11q in defining 2 groups at MM, it should also be used in classifying SMM patients.
3. In addition, the authors have previously shown that MYC rearrangements are highly important in the risk stratification of SMM, but MYC is not included in their classifier. Again, there is an association with MYC and hyperdiploid subgroups and this may be a key factor in determining the subtypes of SMM. The authors should use their MYC data to more accurately define the hyperdiploid SMM samples. They have, in part, done this using expression data, but it is still not clear if MYC rearrangements are the only mechanism of dysregulating MYC expression.
4. There are many ways to classify MM and the authors are performing this here on SMM. However, I find that the new groupings are probably not that different to the ones found in MM, and indeed the authors compare their classification to one of the gene expression classifications used in MM. There seems to be a lot of overlap in Fig 2. Is this classification and nomenclature really any different to previous ones? Or is it just confusing the readers?
5. Given that the authors have performed a classification of SMM and identified high risk groups, how does a physician in the clinic use this information? What is the minimal information required for a physician to be able to use this and stratify their patients? Perhaps high-risk translocations (t(4;14), t(14;16)) and hyperdiploid with MYC rearrangements/high expression?
6. Others have shown that t(14;16) is not high risk in SMM, and indeed this group did not find an association with t(14;16) and high risk of progression in their previous JCO paper. Is this really a high risk group or is it just because it is clustered with the t(4;14)? Many think of the MAF groups as being more stable in the SMM stage and not being high risk at SMM. Can the authors more conclusively convince us that the t(14;16) patients in SMM should be high risk?
7. Raw data should be submitted for the DNA and RNA sequencing performed on these patients to dbGAP, in line with the journals expectations and the groups NIH funding obligations.

Reviewer #2:

Remarks to the Author:

This paper aims to addresses a key clinical question – which patients with smoldering myeloma are at

risk of progression and what are their biological features. Currently the risk scores are based on clinical read-outs such as M component value, BMPC%, SFLC value, and occasionally FISH abnormalities. The authors have used up to 214 cases of smoldering myeloma and characterized mutations and gene expressions to determine 6 distinct subgroups. They have then gone on to compare these subgroups with survival and the current SMM prognostic score.

Recent data in myeloma has suggested that mutational signatures (e.g. apobec, aging, etc) and structural variants (e.g. chromothripsis) may be important in prognosis. Is there any difference in the frequency of these abnormalities between the subgroups and does this link to prognosis in smoldering?

I am unsure what the data concerning the serial clinical features and clusters adds to the story? Is the fall in Hb or rise in M component just a feature of the disease progression and therefore it is no surprise this is more common in the high-risk genetic groups. Or am I missing something?

Please be consistent with the cluster labeling. Sometimes it is C1-6 other times by their initials e.g. FMD, especially in the supplementary data.

All the data should be deposited in dbgap

Reviewer #3:

Remarks to the Author:

Smoldering multiple myeloma is a precursor condition of multiple myeloma which the authors observe to be characterized by significant heterogeneity in terms of disease progression, which is not fully captured by current clinical models; this is the motivation for the need of a more expressive clinical model capable of better characterizing such heterogeneity in disease progression. The authors perform such task by unsupervised binary matrix factorization considering 42 genetic alterations and discover 6 genetic subtypes associated with different clinical features including increased risk of progression to multiple myeloma.

My major worry with this manuscript is the lack of motivation of the algorithm used to perform the analysis and of a comprehensive validation of the results on unseen data. I believe this severely limits any real clinical application of such results.

1) Concerns regarding the methods.

Currently the paper is mostly focused on the description of the translational results. But, I would ask the authors to motivate their computational framework used to determine the 6 clusters. Currently, most of the details are provided as supplementary materials and I failed to fully understand what are all the performed steps (of notice, the link to the Github provided as supplementary material wasn't working) in the analysis.

The authors process different data types (i.e., mutations, copy numbers, etc.) derived from different technologies and settings (i.e., whole exome sequencing with or without matched normal, targeted sequencing, etc.), by means of an elaborate pipeline (which if I understood correctly involves, e.g., mutations correction based on trinucleotide mutational signatures). All of this is briefly described as supplementary materials, but I still think motivations and impact on results should be discussed.

Moreover, I would also ask the authors to motivate and validate their choice of performing clustering based on binary matrix factorization (with many user-defined parameters to tune the final clustering) and how this impacts the final data integration of multiple omic data. Also, a rich literature on data integration to define clusters subtypes is available but totally ignored by the authors, including to name only a few, methods such as Shen, Ronglai, et al. "Integrative subtype discovery in glioblastoma using iCluster." *PloS one* 7.4 (2012): e35236; Ramazzotti, Daniele, et al. "Multi-omic tumor data reveal diversity of molecular mechanisms that correlate with survival." *Nature communications* 9.1

(2018): 1-14. The authors should comment on this and to frame their approach accordingly.

2) Concerns regarding the significance of translational findings.

The authors limit their analysis to showing statistical significance of survival differences of patients, while they should try to validate their results on external datasets; this would be critical in order to prove the translational significance of such results. Ideally, they should find at least one cohort of patients from a different dataset and perform external validation of their method. Furthermore, an additional analysis could be aimed to cross-validate their algorithm by setting aside a random portion of unclustered patients, train the model on remaining patients and show that the algorithm is capable of robustly clustering these unseen patients into different risk groups.

Finally, the authors should demonstrate that their algorithm outperforms standard clinical characteristics in cross-validated survival analysis, to clearly showcase that their expressive model is better than standard predictive metrics used in the clinic.

3) Presentation of results.

Overall, I believe the paper would greatly benefit of thoughtful rewriting and improvement of main text figures in order to provide a clear description of method and results.

REVIEWER COMMENTS

Reviewer #1, expert in MM genomics/transcriptomics (Remarks to the Author):

Here Bustoros et al have performed an analysis on a set of 214 SMM patients to identify genetic subtypes that are associated with the risk of progression to MM. This is a good paper that provides some information on the risk of a SMM patient on the likelihood of progression to MM based on their genomics. Some minor comments are below.

Comments:

1. I find the clustering subtypes a little confusing. It is known that the primary events in myeloma (and its precursors) are immunoglobulin translocations and hyperdiploidy. These primary events are then followed by the addition of copy number changes and mutations. Therefore, I find it surprising that some of the subgroups identified mix together the hyperdiploid and translocation groups – such as the HMC group that contains hyperdiploidy and t(14;20) patients. This doesn't make any sense. It may be that the computational approach puts them together, but etiologically these groups are very different and have separate origins. Previous papers doing similar clustering in the MM stage, using genomics or gene expression, generally always keep the translocation and hyperdiploid groups apart for this very reason. The authors should use some reasoning when deciding the groups and not just follow the informatics.

Response 1: We thank the reviewer for their comments. We want to point out that our clustering analysis was unsupervised and based on only the DNA alterations detected in the 214 samples. We agree with the reviewer that IgH translocations are distinct events from hyperdiploidy. However, we and others (Walker et al., Blood 2018; Maura et al., Nature Comm 2021) have recently reported similar findings, in which t(14;20) or t(14;16) were in the same molecular clusters as Hyperdiploidy. In our cohort, 50% of patients with t(14;20) had hyperdiploidy as well, identified by the presence of trisomies of whole chromosomes and more than 48 chromosomes in the patients' tumor samples.

We clarified this point on pages 6-7 of the updated manuscript, stating, "*The presence of both hyperdiploidy and t(14;20) in the same cluster could be explained by the co-occurrence of those events as described in prior studies (Walker et al., Blood 2018). Indeed, half of patients in our cohort who had t(14;20) also had hyperdiploidy.*"

2. Previous groups looking at MM subgroups have also shown the importance of 11q+ in defining 2 subgroups of hyperdiploidy, which are associated with over-expression of CCND1 and defining the CCDN1/CCND2 expression groups in hyperdiploid MM. The authors do not use this important marker in their classifier. Given the importance of the gain of 11q in defining 2 groups at MM, it should also be used in classifying SMM patients.

Response 2: We thank the reviewer and agree with their comment. We analyzed the four hyperdiploid-like clusters and integrated 11q gain. We found that samples with this copy number gain have higher expression of *CCND1* compared to those without. Furthermore, our analysis showed that *CCND2* expression was significantly higher in samples without 11q gain. However, 11q gain was not a significant feature in any of the hyperdiploid clusters, where 74 out of the 101 tumors had 11q gain. Moreover, we couldn't integrate the *CCND1* or *CCND2* expression data in the disease progression analysis (n=87 patients) because of the small number of samples with RNA data in this subcohort. We updated our text accordingly with these findings on page 8 stating, “Moreover, in the four hyperdiploid clusters, we found that *CCND1* expression was higher in tumors with 11q gain, while *CCND2* expression was higher in samples without 11q gain (Supp Figure 6D-I)”

We also addressed this in the Discussion on pages 13-14, explaining why we could not include *CCND1* and *CCND2* expression groups in the classifier assessing clinical outcomes: “Of note, *CCND1* and *CCND2* expression was analyzed to distinguish between hyperdiploid groups. In our four hyperdiploid groups, we found *CCND1* to be enriched in tumors with 11q gain, while *CCND2* is highly expressed in tumors without 11q gain. We were unable to assess the prognostic impact of this association due to the small number of patients with both gene expression data and clinical follow up for their entire disease course available.”

Figure S6

Supplemental Figure 6: Additional gene expression (log TPM + 1) comparisons. A) *CCND1* in TL2 tumors vs. non-TL2 tumors; B) *CCND2* in TL1 and HL2 tumors vs. non-TL1/HL2 tumors; C) *MCL1* expression in HL1 tumors vs. non-HL1 tumors.

D-I) Comparisons within the hyperdiploid subtypes: HL1, HL2, HL3, HL4. D-F) *CCND1* expression in HL1, HL2, HL3, HL4 subtypes with and without 11q gain, hyperdiploidy, and between these subtypes. G-I) *CCND2* expression in HL1, HL2, HL3, HL4 subtypes with and without 11q gain, hyperdiploidy, and between these subtypes.

3. In addition, the authors have previously shown that MYC rearrangements are highly important in the risk stratification of SMM, but MYC is not included in their classifier. Again, there is an association with MYC and hyperdiploid subgroups and this may be a key factor in determining the subtypes of SMM. The authors should use their MYC data to more accurately define the hyperdiploid SMM samples. They have, in part, done this using expression data, but it is still not clear if MYC rearrangements are the only mechanism of dysregulating MYC expression.

Response 3: We thank the reviewer for their suggestion. Indeed, we and others have previously shown that MYC translocations and amplifications are associated with a higher risk of progression to overt MM. While our first manuscript draft included only MYC amplification (amp 8q24), we repeated the clustering for the study to include MYC translocations. This event is now included in **Figure 1**. MYC translocation is a significant feature in one of the hyperdiploid clusters, hyperdiploid-like (HL) 3, described on page 7, “*Cluster 4 (HL4): this cluster comprises hyperdiploid tumors that are enriched for mutations in KRAS or NFKBIA genes, or MYC translocations as the only significant features.*” Furthermore, MYC copy number amplification (8q24) was also enriched in the hyperdiploid clusters (FDR <0.0001, Fisher’s exact). Transcriptomic data show that MYC expression is significantly higher in the hyperdiploid clusters. All alterations in MYC in our cohort were either translocations or copy number amplifications as stated on page 8: “*MYC oncogene expression was significantly higher in the four hyperdiploid clusters compared to the clusters enriched with IgH translocations (P = 0.009, Wilcoxon Test) (Figure 2D).*”

4. There are many ways to classify MM and the authors are performing this here on SMM. However, I find that the new groupings are probably not that different to the ones found in MM, and indeed the authors compare their classification to one of the gene expression classifications used in MM. There seems to be a lot of overlap in Fig 2. Is this classification and nomenclature really any different to previous ones? Or is it just confusing the readers?

Response 4: Previous studies classified MM subtypes based on gene expression profiling (Zhan et al., Blood, 2006 & Broyl et al., Blood, 2010). Our classification uses only DNA alterations, including translocations, CNAs, and SNVs as the input for binary matrix factorization and consensus clustering to identify molecular subtypes based on co-occurring genetic alterations. Each genetic subtype had significantly enriched genetic

alterations. We identified six clusters, four of which were enriched for hyperdiploid tumors, while two were enriched for known MM IgH translocations. We identified co-occurring features and looked deeply into the molecular makeup of each subtype. We found that combinations of events create phenotypes beyond the effect of a single DNA alteration. For example, hyperdiploidy is generally considered a low-risk feature in MM and SMM; however, we show that not all hyperdiploid tumors are equal, with the HL2 and HL3 clusters having multiple high-risk characteristics in both the transcriptional programs and clinical outcomes. We also identified two tumor clusters enriched for 1q gain: TL1 is enriched in t(4;14) and is clinically high-risk, while HL4 is clinically intermediate risk. In previous expression profile classifications, hyperdiploid (HP) clusters were conflated together and other reported RNA subtypes like low bone (LB), proliferative (PR), and the protein tyrosine phosphatase *PTP4A3* (PRL3) were not mapped to specific cytogenetic or genetic alterations. In our analysis we found that the low bone (LB) disease signature, was upregulated in the HL2, HL4, and TL1 subgroups, suggesting it could be linked to 1q gain, which occurs frequently in these three subgroups, while the PR signature, which is found in proliferative tumor cells, was enriched in the HL3 and TL2 subgroups. The PRL3 signature, which overexpresses the protein tyrosine phosphatase *PTP4A3* and 27 additional genes, was upregulated only in HL4.

We emphasized that our classification is based on the DNA alterations and not the transcriptomic data in the beginning of results section on page 6, *“To identify these patterns, binarized DNA features (42 driver SNVs, CNVs, and translocations) were curated for each sample representing the presence or absence of each genomic alteration. Chapuy et al successfully subtyped diffuse B-cell lymphoma patients with a similar approach, consensus non-negative matrix (NMF) factorization of numeric DNA features. We instead applied consensus BMF for this subtyping to accommodate binarized DNA features, appropriately model summative features that span multiple subtypes (i.e., hyperdiploidy), and handle sparse matrices (Methods).”*

In the Discussion section on page 13: *“We and others have previously cataloged individual driver genetic aberrations in SMM and MM cohorts^{4,11,12}. The present study expands on this work and identifies genetic SMM subtypes defined by multiple recurrent DNA genetic aberrations, whereas previous classification efforts were based primarily on gene expression.”*

We edited the “BMF Clustering Workflow” methods section on pages 16 & 17 for clarity:

“We identified patient subgroups using binarized DNA features and performed consensus binary matrix factorization²⁴. To select the number of clusters (K) for the consensus clustering, we randomly downsampled our input matrix and computed silhouette scores using Dice dissimilarity, residuals of factorization fit, variance explained, and K-L divergence on binary matrix factorizations over a range of K. We found a decrease in K-L divergence with our full dataset from K = 5 to K = 6, which suggested that 6 clusters were best suited to ensure a converged factorization for N = 214 (Supp. Fig 2) Additionally, we found that variance stabilized when we performed down sampling analyses at N = 75-

100, suggesting we were powered to perform binary matrix factorization for a cohort at this minimum size. We concluded that a minimum of 100 samples and 6 clusters were suited for this approach. We performed consensus clustering using a binary matrix factorization with K of 2 through 10, selecting the final 6 clusters based on hierarchical clustering of the consensus matrix with Euclidean distance and Ward linkage. We assessed binary feature importance by performing a Fisher's exact test to count feature representation within each cluster and outside of this cluster, testing for an equal proportion. The false discovery rate (FDR) was calculated using the Benjamini-Hochberg procedure."

5. Given that the authors have performed a classification of SMM and identified high risk groups, how does a physician in the clinic use this information? What is the minimal information required for a physician to be able to use this and stratify their patients? Perhaps high-risk translocations (t(4;14), t(14;16)) and hyperdiploid with MYC rearrangements/high expression?

Response 5: We thank the reviewer for their comment and hope to clarify how we envision a clinician using these subtypes.

MM genetic alterations do not occur individually but together, and we aimed to detect and define the patterns of co-occurrence in MM patients. Each of the six genetic subtypes we identified and their significantly enriched features are summarized in **Table 1** below and **Figure 1** in the main manuscript to help readers and clinicians identify the most important genetic features and associated clinical risk.

We defined the significant genetic alterations occurring in the high-risk clusters in the results section on Page 11 and listed the significant features in the supplementary data: "We found that high-risk genetic subtypes were significantly enriched with specific genetic alterations, such as, *KRAS*, *TP53*, t(4;14), 1q gain, and 16q, 8p, and 1p deletions among others (**Supp. Table 3**). Of note, many of these high-risk genetic events were associated with higher risk of progression to MM (**Supp. Table 5 & 6**).

Moreover, to validate our findings on external cohorts and allow for future classification of SMM patients into our subtypes, we trained a Random Forest (RF) classifier on our cohort of 214 samples. We applied this classifier to two external cohorts (**Fig 3F**) and validated the survival effect described previously. To interrogate what features were most important for the classifier, we considered feature importance for the RF training process (**Supp. Fig 5C**). We determined the most pertinent features for subtype classification are: hyperdiploidy, 13q deletion, t(11;14), t(4;14), 16q deletion, 1q Amp, *KRAS* mutations, 6q deletion, and 14q deletion.

Genomic Subtypes of SMM (n=214)

	Name	Risk	SNVs	CNVs	Translocations
C1	HL1	Intermediate	NRAS, TRAF3, MAX	HP	
C2	HL2	High	MAFB, NRAS, BRAF, TP53, ATM	HP, 16q Del [CYLD]	t(14;20)
C3	TL1	High	MAF, DIS3, HIST1H1E, PRKD2, BRAF	14q Del, 13q Del	t(4;14) [FGFR3/MMSET], t(14;16) [MAF]
C4	HL3	High	KRAS, NFKBIA	HP	MYC
C5	TL2	Low		11q Amp	t(11;14) [CCND1]
C6	HL4	Intermediate	NFKB2	HP, 2p Amp	

Figure S5

Supplemental Figure 5: **A)** Random forest K-fold cross validation accuracy over 10 repeats with mean accuracy and standard deviation reported. **B)** Grid search to optimize parameters did not improve over initial random forest parameters. **C)** The prevalence of each feature in the dataset plotted by feature importance in classifying genetic subtypes.

6. Others have shown that t(14;16) is not high risk in SMM, and indeed this group did not find an association with t(14;16) and high risk of progression in their previous JCO paper. Is this really a high risk group or is it just because it is clustered with the t(4;14)? Many think of the MAF groups as being more stable in the SMM stage and not being high risk at SMM. Can the authors more conclusively convince us that the t(14;16) patients in SMM should be at high risk?

Response 6: We thank the reviewer for bringing up this point. Thus far, our group and others have not seen t(14;16) as a high risk feature in progression to active MM. However, the underrepresentation of t(14;16) in our and other cohorts limited the statistical power to identify an association with outcome. Here, our combined primary and validation cohorts, which increased the total number of patients with available disease course data from 87 to 229, had only 6 patients (3%) with the t(14;16) translocation. The Kaplan-Meier figure below for time to progression (TTP) showed a signal of higher risk of t(14;16) patients, but the difference was not statistically significant. We also found that t(14;16) was associated with other high-risk features like *TP53* mutations and deletions ($p = 0.007$) and APOBEC mutational signature ($p = 0.005$). (Bustoros et al. JCO, 2020). Although we do not have definitive evidence for its association with increased risk of progression, we cannot rule it out. Larger cohorts enriched for t(14;16) may be needed to answer this question.

On page 15, we added, "We and others have not found them to confer a high risk of progression on their own (add reference here for others). However, multiple studies have shown that t(14;16) is frequently associated with APOBEC signature and genomic instability⁴⁻¹⁰. In our study it was found in 5% of patients and with similar rates in the validation cohorts, so larger studies with cohorts enriched for t(14;16) may be needed to confidently determine their prognostic significance in SMM."

7. Raw data should be submitted for the DNA and RNA sequencing performed on these patients to dbGAP, in line with the journals expectations and the groups NIH funding obligations.

Reply: We would like to thank the reviewer for pointing this out. We have deposited the data underlying this study to dbGaP. We have added a statement to document this in the main manuscript on page 18, *“The DNA and RNA sequencing data and analyses presented in the current publication have been deposited in and are available from the dbGaP database under dbGaP accession phs001323.v2 p1.”*

Reviewer #2, expert in MM/SM subtypes (Remarks to the Author):

This paper aims to address a key clinical question – which patients with smoldering myeloma are at risk of progression and what are their biological features. Currently the risk scores are based on clinical read-outs such as M component value, BMPC%, SFLC value, and occasionally FISH abnormalities. The authors have used up to 214 cases of smoldering myeloma and characterized mutations and gene expressions to determine 6 distinct subgroups. They have then gone on to compare these subgroups with survival and the current SMM prognostic score.

Recent data in myeloma has suggested that mutational signatures (e.g., APOBEC, aging, etc) and structural variants (e.g., chromothripsis) may be important in prognosis. Is there any difference in the frequency of these abnormalities between the subgroups and does this link to prognosis in smoldering?

Reply: We thank the reviewer for their comment. To address this, we used mutational signatures derived from 72 samples with WES using SignatureAnalyzer (<https://github.com/broadinstitute/getzlab-SignatureAnalyzer>). We found that APOBEC mutational signature activity (SBS 2 & SBS13) differed significantly between the six subtypes ($P = 0.027$, Kruskal-Wallis), while AID mutational signatures did not ($P = 0.17$, Kruskal-Wallis) (**Supp. Fig. 1F, G**). Specifically, we found that APOBEC activity is enriched in the HL2 and TL1 (high-risk) clusters vs. the rest ($P = 0.006$, Wilcoxon test, **Supp. Fig 1H**) confirming that these tumor subtypes harbor multiple high-risk features.

We updated this in the main text on page 9: “We asked whether these genetic subtypes were enriched for specific mutational signatures, and found that the APOBEC mutational signature activity (SBS 2,13 COSMIC 3) differed between the genetic subtypes ($P = 0.027$, Kruskal-Wallis) while AID mutational signatures did not ($P = 0.17$) (Supp. Fig. 1F, G). Specifically, we found that the APOBEC activity signature was enriched in the HL2

and TL1 clusters compared to the rest of tumors (P = 0.006, Supp. Fig 1H), providing further evidence of the high-risk prognostic genetic biomarkers in these two subtypes.”

I am unsure what the data concerning the serial clinical features and clusters adds to the story? Is the fall in Hb or rise in M component just a feature of the disease progression and therefore it is no surprise this is more common in the high-risk genetic groups? Or am I missing something?

Reply: We thank and agree with the reviewer. Hb or M component dynamic changes and trajectories are important measures to assess patient risk of progression. We wanted to illustrate that the genetic subtypes are predictive of these serial changes regardless of progression status. However, to avoid confusion, we have removed this data from the current manuscript as this data requires additional discussion and context.

Please be consistent with the cluster labeling. Sometimes it is C1-6 other times by their initials e.g. FMD, especially in the supplementary data.

Reply: We thank the reviewer for their comment. We have simplified the cluster/subtype names to hyperdiploid-like (1-4) and translocation-like (1-2) to include the subtype name throughout the manuscript and supplemental data. Additionally, we provide a table with significant features for each subtype (Fig 1C) for ease of reference.

All the data should be deposited in dbgap.

Reply: We have now deposited the appropriate data in dbGaP. We also added a statement to document this in the methods section: *“The DNA and RNA sequencing data and analyses presented in the current publication have been deposited in and are available from the dbGaP database under dbGaP accession phs001323.v2.p1.”*

Reviewer #3, expert in bioinformatics for subtype classification (Remarks to the Author):

Smoldering multiple myeloma is a precursor condition of multiple myeloma which the authors observe to be characterized by significant heterogeneity in terms of disease progression, which is not fully captured by current clinical models; this is the motivation for the need of a more expressive clinical model capable of better characterizing such heterogeneity in disease progression. The authors perform such task by unsupervised binary matrix factorization considering 42 genetic alterations and discover 6 genetic subtypes associated with different clinical features including increased risk of progression to multiple myeloma.

My major worry with this manuscript is the lack of motivation of the algorithm used to perform the analysis and of a comprehensive validation of the results on unseen data. I believe this severely limits any real clinical application of such results.

1) Concerns regarding the methods.

Currently the paper is mostly focused on the description of the translational results. But, I would ask the authors to motivate their computational framework used to determine the 6 clusters. Currently, most of the details are provided as supplementary materials and I failed to fully understand what are all the performed steps (of notice, the link to the Github provided as supplementary material wasn't working) in the analysis.

The authors process different data types (i.e., mutations, copy numbers, etc.) derived from different technologies and settings (i.e., whole exome sequencing with or without matched normal, targeted sequencing, etc.), by means of an elaborate pipeline (which if I understood correctly involves, e.g., mutations correction based of trinucleotide mutational signatures). All of this is briefly described as supplementary materials, but I still think motivations and impact on results should be discussed.

Moreover, I would also ask the authors to motivate and validate their choice of performing clustering based on binary matrix factorization (with many user-defined parameters to tune the final clustering) and how this impacts the final data integration of multiple omic data. Also, a rich literature on data integration to define clusters subtypes is available but totally ignored by the authors, including to name only a few, methods such as Shen, Ronglai, et al. "Integrative subtype discovery in glioblastoma using iCluster." PloS one 7.4 (2012): e35236; Ramazzotti, Daniele, et al. "Multi-omic tumor data reveal diversity of molecular mechanisms that correlate with survival." Nature communications 9.1 (2018): 1-14. The authors should comment on this and to frame their approach accordingly.

We thank the reviewer for this comment. First, the Github link (https://github.com/getzlab/SMM_clustering_2020) is now public and provides every

computational analysis performed in annotated Jupyter notebooks. This is now referenced in the **Methods** on page 18, under **Code Availability**.

Our computational workflow for subtyping was first motivated by the available input patient data. The genomic data for our primary cohort consists of tumor-normal sample pairs with whole-exome sequencing (WES) or tumor-only samples with targeted sequencing to identify SNVs and CNVs. FISH was used to identify translocations. Additional information about data processing can be found in our previous publication¹. We added this reference to the Methods section as well.

Previous studies have carefully outlined the importance of driver mutations, CNVs, and translocations in the progression of myeloma. However, many of these measurements are coarsely described as the presence or absence of a modifying event thresholded by statistical significance (e.g., the presence of a KRAS driver mutation in a patient). Many clinical data similarly report the presence or absence of important genomic events (i.e., FISH for translocations) that are useful for guiding clinical decisions. We decided to subtype patients using binarized SNVs, CNVs, and translocations easily standardized in different clinical contexts. Recently, Chapuy *et al* successfully used numerically encoded DNA genomic data to identify clinically relevant subtypes of diffuse large B-cell lymphoma with consensus non-negative matrix factorization (NMF) of concatenated, multi-omic features.² NMF is widely used as a robust clustering method for sparse data³⁴ We take a similar approach but constrain matrix factorization to Bernoulli distributed features (binary matrix factorization, BMF) for the binarized nature of this dataset.

We thank the reviewer for the comment about user-defined hyperparameters. We performed a downsampling analysis to 1) identify the requisite number of samples for stable clustering (i.e., power analysis) and 2) identify the number of clusters patients stably group into. As discussed in the original methods, we use variance explained and K-L divergence when downsampling our primary cohort (n = 214) 100 times for binary matrix factorization set to a range of hyperparameter K=2 to 10 (**Supp Fig 2A-E**). These results suggest that 100 samples are necessary for stable clustering and to derive 6 groups or clusters of samples. We performed consensus clustering using a binary matrix factorization with K of 2 through 10, selecting the final 6 clusters based on hierarchical clustering of the consensus matrix with Euclidean distance and Ward linkage. We assessed binary feature importance by performing a Fisher's exact test to count feature representation within each cluster and outside of this cluster, testing for an equal

¹ Bustoros, Mark, et al. "Genomic profiling of smoldering multiple myeloma identifies patients at a high risk of disease progression." *Journal of Clinical Oncology* 38.21 (2020): 2380.

² Chapuy, Bjoern, et al. "Molecular subtypes of diffuse large B cell lymphoma are associated with distinct pathogenic mechanisms and outcomes." *Nature medicine* 24.5 (2018): 679-690.

³ Alexandrov, L. B., Kim, J., Haradhvala, N. J., Huang, M. N., Ng, A. W. T., Wu, Y., ... & Islam, S. A. (2020). The repertoire of mutational signatures in human cancer. *Nature*, 578(7793), 94-101.

⁴ Zavidij, Oksana, et al. "Single-cell RNA sequencing reveals compromised immune microenvironment in precursor stages of multiple myeloma." *Nature cancer* 1.5 (2020): 493-506.

proportion. The false discovery rate (FDR) was calculated using the Benjamini-Hochberg procedure.

Many multi-omic and multi-view clustering approaches, including the two sources provided by the reviewer, rely on heuristics to select an appropriate number of subtypes.

There are, indeed, a vast number of approaches used for clustering of multi-omic data. *Rappoport et al*⁵ consider an approach like ours an “early integration” approach, whereby we concatenate separate “omic” features and perform clustering on combined feature space. “Late integration” approaches cluster data separately for each data type and then integrate these results, such as COCA⁶ and PINS⁷. These are useful for unifying independent clustering done on assays with different distributional assumptions to cohesively subtype patients. However, our featurization binarizes genomic events for all three modalities (SNVs, CNVs, translocations), and allows us to use a similar clustering approach for each these “omic” datasets. Further, we have ≤ 6 translocations available due to limitations in clinical FISH measurements, which renders a “late integration” approach difficult due to an insufficient number of features. Dimensionality reduction algorithms like jointNMF⁸ and multiNMF⁹ that perform reduction on each data type separately are similarly difficult to apply to our translocation feature space ($m=6$). Thus, identifying an algorithm that can account for translocations, an important alteration in the pathogenesis of multiple myeloma, is crucial.

The reviewer points out a probabilistic approach, iCluster¹⁰, that computes a low dimensionality composition of each data modality with noise modeled explicitly. It uses an EM-like algorithm and then K-means on the lower dimension to suggest likely sample cluster groupings. iCluster is designed to fit Gaussian distributed data. While we could z-score CNVs per chromosomal arm using the number of copies, this is not possible for FISH translocation data because these assays detect the presence or absence of an event. Additionally, methods like iCluster are particularly equipped to handle feature selection for high dimensionality “omic” data, such as gene expression. For this study, we only had RNA expression data for a subset of patients, which is why transcriptomics is not included in clustering of the primary cohort. Finally, the reviewer commented on the use of hyperparameters: iCluster is parameterized by the number of subtypes expected, K, and its authors suggest using a cluster reproducibility index (RI) as a heuristic for selecting this value while partitioning samples into learning and test sets. We uses K-L divergence

⁵ Rappoport, Nimrod, and Ron Shamir. "Multi-omic and multi-view clustering algorithms: review and cancer benchmark." *Nucleic acids research* 46.20 (2018): 10546-10562.

⁶ Hoadley K.A., Yau C., Wolf D.M., Cherniack A.D., Tamborero D., Ng S., Leiserson M.D., NiuB., McLellan M.D., Uzunangelov V. et al. Multiplatform analysis of 12 cancer types reveals molecular classification within and across tissues of origin. *Cell*. 2014; 158:929–944.

⁷ Nguyen T., Tagett R., Diaz D., Draghici S. A novel approach for data integration and disease subtyping. *Genome Res*. 2017; 27:2025–2039.

⁸ Zhang S., Liu C.-C., Li W., Shen H., Laird P.W., Zhou X.J. Discovery of multi-dimensional modules by integrative analysis of cancer genomic data. *Nucleic Acids Res*. 2012; 40:9379–9391.

⁹ Liu J., Wang C., Gao J., Han J. Multi-View Clustering via Joint Nonnegative Matrix Factorization. *Proc. ICDM '13*. 2013; Philadelphia, PASociety for Industrial and Applied Mathematics 252–260.

¹⁰ Shen, Ronglai, et al. "Integrative subtype discovery in glioblastoma using iCluster." *PLoS one* 7.4 (2012): e35236.

and explained variance in random downsampling to ensure the robustness of the results—both methods require heuristic approaches to select K. Bayesian non-parametric models (e.g., the Chinese Restaurant Process¹¹) could potentially be used in future analyses to avoid the pitfalls of these approaches for multi-omic data.

Patient similarity-based approaches for clustering, such as similarity network fusion (SNF)^{12,13}, have recently shown promise in large-scale multiple myeloma cohorts¹⁴. SNF constructs similarity networks for each omic separately and proceeds to fuse modalities based on an iterative approach. The reviewer points to newer a approach by Ramazzotti *et al.*, Cancer Integration via Multi Kernel Learning (CIMLR)¹⁵, to perform integrated analysis of multi-omic data. CIMLR learns pairwise similarity metrics by combining gaussian kernels per “omic.” This results in a similarity matrix that is used for dimensionality reduction and *k-means* clustering. Thus, CIMLR additionally has a cluster selection step that requires iterating over a list of values of K based on separation cost to multi-omics. Ramazzotti *et al* use mutation data (binarized) as its input and subsequently normalize data so that values range between 0 and 1 for data types, including transcription, CNVs, and DNA methylation. For this cohort, such an approach was not needed to integrate different data types since all features were binarized. A future formulation of CIMLR that does not require fitting Gaussian kernels for data types but rather is formulated around Bernoulli distributed data would potentially better fit the need of this SMM dataset. Our NMF approach is suited for sparse, non-negative, binarized data and cluster stability is described with accompanying downsampling analysis. This approach is well suited for the data, disease biology, and potential applications of the present study.

This is one of the largest cohorts with genomic data of Smoldering Multiple Myeloma patients clinically defined at this precursor stage. As more patient cohorts are gathered with additional sequencing assays (e.g., transcriptomics, miRNA, proteomics, chromatin accessibility), alternative approaches for multi-omics/multi-view clustering with continuous valued data would be a natural next step to understanding the risk of these patients.

2) Concerns regarding the significance of translational findings.

The authors limit their analysis to showing statistical significance of survival differences of patients, while they should try to validate their results on external datasets; this would be

¹¹ Blei, David M., Thomas L. Griffiths, and Michael I. Jordan. "The nested chinese restaurant process and bayesian nonparametric inference of topic hierarchies." *Journal of the ACM (JACM)* 57.2 (2010): 1-30.

¹² Bo Wang, Jiayan Jiang, Wei Wang, Zhi-Hua Zhou, Zhuowen Tu Unsupervised metric fusion by cross diffusion. *2012 IEEE Conference on Computer Vision and Pattern Recognition*. 2012; IEEE2997–3004.

¹³ Wang, Bo, et al. "Similarity network fusion for aggregating data types on a genomic scale." *Nature methods* 11.3 (2014): 333-337.

¹⁴ Bhalla, Sherry, et al. "Patient Similarity Network of Newly Diagnosed Multiple Myeloma Identifies Patient Sub-groups with Distinct Genetic Features and Clinical Implications." (2020).

¹⁵ Ramazzotti, Daniele, et al. "Multi-omic tumor data reveal diversity of molecular mechanisms that correlate with survival." *Nature communications* 9.1 (2018): 1-14.

critical in order to prove the translational significance of such results. Ideally, they should find at least one cohort of patients from a different dataset and perform external validation of their method. Furthermore, an additional analysis could be aimed to cross-validate their algorithm by setting aside a random portion of unclustered patients, train the model on remaining patients and show that the algorithm is capable of robustly clustering these unseen patients into different risk groups.

We thank the Reviewer for their comment and agree completely. To address this, we have included two independent validation cohorts. First, we trained a Random Forest Classifier and obtained a mean, 5-fold CV training accuracy of 86.7% (+/- 5%) using our 214 samples and 36 binary features. We then applied this classifier to two cohorts and validate the survival effect. We used an external cohort of 75 SMM patients to validate the classifier and investigate whether the genetic subtypes were predictive of progression¹¹. The patients in this cohort were enriched in the low-risk clinical group and had a median time to progression (TTP) of 5 years. Like the primary cohort, patients in the intermediate and high-risk genetic subtypes had increased risk of progression to active MM in multivariate analysis accounting for the clinical risk group (HR: 4.5 and 9, $P = 0.039$ and 0.002 , respectively) (**Fig A**). We found that adding the genetic risk groups improved the prediction of progression compared to the clinical-only model (C-index: 0.76 vs 0.65). We use an independent, smaller cohort of 67 patients with targeted capture data, including common MM translocations, CNAs, and SNVs, to extend our validation cohort. In the combined validation cohort of 142 patients, being assigned to the TL1, HL2, HL3, or HL4 subtypes were independent predictors of progression to active myeloma (**Fig B**). The high-risk genetic subtypes (HL2, HL3, TL1) were associated with increased risk of progression in multivariate analysis (HR: 3.4, 95% CI :1.68-6.7). We then asked whether combining the three cohorts would provide more power to test the clinical significance of our genetic classification. Furthermore, to increase the power of our analysis, we combined the three cohorts and found the same effect with greater significance compared to our initial findings. The low, intermediate, and high genomic risk groups had a different TTP (**Fig C**), and the high-risk genetic subtypes had significantly shorter TTP compared to the low or the intermediate risk groups (**Fig D**). We also found that both the individual genetic subtypes and the genetic risk groups were independent predictors of progression in the combined cohort multivariate analysis, validating our initial findings (**Fig E and F**).

A)

B)

C)

D)

E)

F)

Figure legends

A) Multivariate cox regression analysis of the clinical risk stages and the genetic in the first validation cohorts of 75 patients **(B)** Multivariate cox regression analysis of the clinical risk stages and the genetic risk groups according to the IMWG 20/2/20 model in the two validation cohorts of 75 patients and 67 patients. **C)** Kaplan-Meier curves for analysis of TTP in patients belonging to the three genetic risk groups of the combined cohort. **D)** Kaplan-Meier curves for analysis of TTP in patients from the 6 genetic subtypes in the combined cohort of 229 patients **E)** Multivariate cox regression analysis of the low, intermediate, and high-risk genetic subtypes and clinical risk stages according to the IMWG 20/2/20 model in the primary cohort. **F)** Multivariate cox regression analysis of the low, intermediate, and high-risk genetic subtypes and clinical risk stages according to the IMWG 20/2/20 model in the combined cohorts.

The new added paragraph of the results of the validation cohorts are written in the results section on page 11 &12:

“Validation of the molecular subtypes in external cohorts

*To validate our findings on the clinical significance of the genetic subtypes, we developed a classifier based on the features of the clusters we identified in our primary cohort. We used an external cohort of 75 SMM patients to validate the classifier and investigate whether the genetic subtypes are predictive for progression¹¹. The patients in this cohort were enriched in the low-risk clinical stage and had a median TTP of 5 years. Like the primary cohort, patients in the intermediate and high-risk genetic subtypes had increased risk of progression to active MM in multivariate analysis accounting for the clinical risk stage (HR: 4.5 and 9, P = 0.039 and 0.002, respectively) (**Figure 3D**). We found that adding the genetic risk groups improved the prediction of progression compared to the clinical model only (C-index: 0.76 vs 0.65, respectively). We also obtained another smaller cohort of 67 patients with targeted capture data, including common MM translocations, CNAs and SNVs, and added it to the previous cohort (cite). In those 142 patients, HL2, HL3, HL4, and TL1 subtypes were independent predictors of progression to active myeloma (**Figure 3E**) and the high-risk genetic subtypes were associated with increased risk of progression in multivariate analysis (HR: 3.4, 95% CI :1.68-6.7) (**Supp. Figure 10**). We then asked whether combining all the three cohorts would provide more power to test the clinical significance of our genetic classification. The combined cohort contained 229 SMM patients with median follow-up and TTP of 7 and 5 years, respectively. Indeed, the genetic subtypes had a different TTP (**Figure 3F**), and the high-risk genetic subtypes had significantly shorter TTP compared to the low or the intermediate risk groups (**Figure 3G**). We also found that both the individual genetic subtypes and the genetic risk groups were independent predictors of progression in the combined cohort multivariate analysis, validating our initial findings (**Figure 3H & Supp. Figure 11**).”*

Finally, the authors should demonstrate that their algorithm outperforms standard clinical characteristics in cross-validated survival analysis, to clearly showcase that their expressive model is better than standard predictive metrics used in the clinic.

We thank the reviewer for their comment. As described above, when validating our classifier's results on the two independent validation cohorts, we also stratified patients using the current standard-of-care IMWG clinical risk model. We used the C-index statistic to assess the performance of the clinical vs the clinical and genetic models in predicting progression in both the primary, validation, and the combined cohorts and updated the text accordingly. This is described in the Results, under the subheading **Validation of the molecular subtypes in external cohorts**, on page 11-12 (quoted above).

3) Presentation of results.

Overall, I believe the paper would greatly benefit of thoughtful rewriting and improvement of main text figures in order to provide a clear description of method and results.

We thank the reviewer for their comment. We have rewritten the main text figures and methods section of the paper. Of note, when introducing the methodology used for patient subtyping, we provide the rationale for the approach taken in **Results** on page 6, *“To identify these patterns, binarized DNA features (42 driver SNVs, CNVs, and translocations) were curated for each sample representing the presence or absence of each genomic alteration. Chapuy et al successfully subtyped diffuse B-cell lymphoma patients with consensus non-negative matrix (NMF) factorization of numeric DNA features. We instead apply consensus BMF for this multi-omics subtyping to accommodate these binarized DNA features, appropriately model summative features that span multiple subtypes (i.e., hyperdiploidy), and handle sparse matrices (**Methods**).”*

The precise steps taken for subtyping were clarified in **Methods** on page 17, *“We performed consensus clustering using a binary matrix factorization with K of 2 through 10, selecting the final 6 clusters based on hierarchical clustering of the consensus matrix with Euclidean distance and Ward linkage. We assessed binary feature importance by performing a Fisher's exact test to count feature representation within each cluster and outside of this cluster, testing for an equal proportion. The false discovery rate (FDR) was calculated using the Benjamini-Hochberg procedure.”*

Reviewers' Comments:

Reviewer #3:

Remarks to the Author:

I believe this revised version of the manuscript to be greatly improved. In particular, I appreciate that the authors have now included a validation in external cohort section, which in my opinion provides much better support to the results.

Overall, I think they provide reasonable justification for their methodology in the answer to reviewers. However, I would ask the authors to also include a short description of such motivations in the main text (in the methods section or where they think it best fits) and cite/comment other possible approaches (such as iCluster and CIMLR) for subtyping as they have already done answering my concern #1.

Reviewer #4:

Remarks to the Author:

Reviewer '1B' comments on revised manuscript and authors' response to Reviewer 1

Overall:

I agree this is a good paper, with the extensive revisions showing the predictive benefits of the DNA alteration-based clusters in smouldering myeloma, now also in validation cohorts. Their relationship with well-known gene expression profile-defined myeloma subgroups is interesting. It would benefit from improved figure referencing as per comments below.

Responses to authors' rebuttals to reviewer 1 comments:

1. I have no problem with this, and agree there are cases with HRD and t(14;20). However please note that the extra sentence now added in the manuscript page 6:

'The presence of both hyperdiploidy and t(14;20) in the same cluster could be explained by either the small number of samples with these alterations as seen in other studies as well, or that in few cases they co-occur together. Indeed, half of patients with t(14;20) in our cohort had hyperdiploidy.'

Is a little clumsy and not the same as the sentence quoted in your rebuttal

"The presence of both hyperdiploidy and t(14;20) in the same cluster could be explained by the co-occurrence of those events as described in prior studies (Walker et al., Blood 2018). Indeed, half of patients in our cohort who had t(14;20) also had hyperdiploidy."

Which is clearer.

2. No additional comments - thoroughly discussed in revised manuscript.

Note: Supp Figure S6 is not referenced at all in the main text.

3. Although I can't really read the genetic event y-axis labels in Fig 1B, the number of MYC translocations identified in the whole dataset as per Fig 1B is underwhelming (looks like n=2 in HL3 and n=2 in TL1)? While MYC TLs are reported in text to be 'significant' in cluster 4/HL3 (note I cannot see any evidence for this significance, cannot find in supplements), do 2 cases really convince as enrichment in HL3 as suggested? May some MYC complexity be missed by WES/TRS approaches used? 4/214 is much fewer cases of MYC TL that would be expected in a symptomatic myeloma dataset, and although this is a smouldering dataset, this low number should be discussed. I note that when you report total MYC aberrations (which presumably includes CNAs), you report incidence of 7/87 9% in primary dataset, which increases to 21% on combination with your validation cohorts, highlighting the unusual lack of MYC aberrations in your dataset but presence in validation cohorts. For comparison: <https://www.ncbi.nlm.nih.gov/pmc/articles/PMC6923575/> - 24% SMM had MYC SVs on targeted sequencing.

<https://www.nature.com/articles/s41467-020-20524-2> - (validation dataset used here) 35% SMM had MYC rearrangements on targeted sequencing.

<https://www.nature.com/articles/s41467-018-05058-y> - WGS - 5/11 cases showed translocations of MYC

4. I agree with reviewer 1 that biologically, findings are compatible with previous work identifying high risk in myeloma, rather than identifying novel prognostic biology, and generally support others' findings that SMM is really genetically very similar to symptomatic MM.

However, I agree with authors that a DNA-only based classifier is helpful as this is more accessible to most clinicians than RNASeq. Nonetheless it could still be clearer for the non-expert reader, where BMF approach is described early in results, that the transcriptomic data does not contribute to the clustering of the primary cohort, particularly as it is referred to as 'multi-omics' subtyping: "We instead apply consensus BMF for this multi-omics subtyping" - this is a bit misleading.

5. As above, I find the concept that clustering was done using DNA panel-detected genomic events helpful, as this is more accessible to most clinicians than RNASeq, in contrast to previously-published gene-expression-based clustering, as authors point out. Fig 1C is helpful to this end, as is the determination of the minimum classifiers required.

Supp Fig 5 not referenced in the text?

6. I am happy with this answer and agree the t(14;16) cohort is too small for conclusive association analysis.

7. Fine

Extra small comments

1. Where clusters are described on page 7, the sentence 'We named this cluster Translocation-like1 (TL1).' Is used in reference to cluster 3 AND cluster 5 - typo.

2. Gene names Fig 1B are poorly legible

3. There seems to be an error where Fig. 2 has been updated but the text referring to it has not been? 2A-E references in the main text do not match and the quoted p values do not match the figure? Also confusion between 'HP' and 'HL' label in legend. 2F/G called 'FMD' and 'CND' in legend but 'TL1' and 'TL2' on the figure is confusing. There are 2 'G's.

4. Even enlarging 500% on my screen I cannot read the gene names in 2F and G!

5. Newly added sentence about supp table 4 (mis-referenced as 3) - this table suddenly introduces extra validation cohorts before the appropriate text section explaining them, which is confusing

6. Several of the supp figures are not referenced in the text / not ordered as per reference in text.

Reviewer #5:

Remarks to the Author:

The concerns were addressed satisfactorily. However, it is unclear whether the P values provided for the significance of the APOBEC activity signature were adjusted for multiple testing. If not, the authors should include the adjusted data.

Responses to the second round of reviewers' comments

Reviewer #3 (Remarks to the Author):

I believe this revised version of the manuscript to be greatly improved. In particular, I appreciate that the authors have now included a validation in external cohort section, which in my opinion provides much better support to the results.

Overall, I think they provide reasonable justification for their methodology in the answer to reviewers. However, I would ask the authors to also include a short description of such motivations in the main text (in the methods section or where they think it best fits) and cite/comment other possible approaches (such as iCluster and CIMLR) for subtyping as they have already done answering my concern #1.

We thank the reviewer for their comment. We added a whole paragraph in the Methods section (page 19) where we discussed the clustering approaches for different data types and what we used in our current studies as follows:

“Clustering Approach. A vast number of approaches have been applied to clustering multi-omic sequencing data. Late integration algorithms that cluster data types separately and then integrate them to a final result, such as COCA and PINS were previously described⁴⁰⁻⁴². Dimensionality reduction algorithms such as jointNMF and multiNMF similarly factorize each data type separately before final integration^{43,44}. iCluster is a probabilistic approach that computes a low dimensionality composition of each data modality designed to fit Gaussian distributed data⁴⁵. However, our translocation measurement includes 6 sparse events, rendering any approaches requiring clustering of this data type separately or with Gaussian data assumptions difficult to apply. Furthermore, our choice of using binarized features (SNVs, CNVs, and translocations) allows us to use a simpler, “early integration” approach, where the feature space is combined before the algorithm is applied. Future work using patient similarity-based approaches, such as similarity network fusion (SNF) or Cancer Integration via Multi

Kernel Learning (CIMLR) are promising future directions^{46,47}. Formulating these to account for Bernoulli distributed translocation data would be ideal for including these important drivers of MM pathogenesis. As more SMM patient cohorts are gathered and sequenced with transcriptomic, proteomic, and chromatin accessibility data, applying patient similarity approaches and algorithms with more appropriate distributional assumptions is key. However, our BMF approach is appropriate for sparse, non-negative, binarized data curated in this initial cohort.”

Reviewer #4

assessed responses to Reviewer's #1 previous requests (Remarks to the Author):

Reviewer '1B' comments on revised manuscript and authors' response to Reviewer 1

Overall:

I agree this is a good paper, with the extensive revisions showing the predictive benefits of the DNA alteration-based clusters in smouldering myeloma, now also in validation cohorts. Their relationship with well-known gene expression profile-defined myeloma subgroups is interesting. It would benefit from improved figure referencing as per comments below.

Responses to authors' rebuttals to reviewer 1 comments:

1. I have no problem with this, and agree there are cases with HRD and t(14;20). However please note that the extra sentence now added in the manuscript page 6:

'The presence of both hyperdiploidy and t(14;20) in the same cluster could be explained by either the small number of samples with these alterations as seen in other studies as well, or that in few cases they co-occur together. Indeed, half of patients with t(14;20) in our cohort had hyperdiploidy.'

Is a little clumsy and not the same as the sentence quoted in your rebuttal

“The presence of both hyperdiploidy and t(14;20) in the same cluster could be explained by the co-occurrence of those events as described in prior studies (Walker et al., Blood 2018). Indeed, half of patients in our cohort who had t(14;20) also had hyperdiploidy.”

Which is clearer.

- We thank the reviewer for their feedback, and we agree with their comments. We updated the text as highlighted by the reviewer.

2. No additional comments - thoroughly discussed in revised manuscript.

Note: Supp Figure S6 is not referenced at all in the main text.

- We added the reference for Supp Fig 6 in the main text. We changed the order of the supplementary figures, so it is now cited in page 8, as Supplementary Figure 2.

3. Although I can't really read the genetic event y-axis labels in Fig 1B, the number of MYC translocations identified in the whole dataset as per Fig 1B is underwhelming (looks like n=2 in HL3 and n=2 in TL1)? While MYC TLs are reported in text to be 'significant' in cluster 4/HL3 (note I cannot see any evidence for this significance, cannot find in supplements), do 2 cases really convince as enrichment in HL3 as suggested? May some MYC complexity be missed by WES/TRS approaches used? 4/214 is much fewer cases of MYC TL that would be expected in a symptomatic myeloma dataset, and although this is a smouldering dataset, this low number should be discussed. I note that when you report total MYC aberrations (which presumably includes CNAs), you report incidence of 7/87 9% in primary dataset, which increases to 21% on combination with your validation cohorts, highlighting the unusual lack of MYC aberrations in your dataset but presence in validation cohorts. For comparison:

<https://www.ncbi.nlm.nih.gov/pmc/articles/PMC6923575/> - 24% SMM had MYC SVs on targeted sequencing.

<https://www.nature.com/articles/s41467-020-20524-2> - (validation dataset used here) 35% SMM had MYC rearrangements on targeted sequencing.

<https://www.nature.com/articles/s41467-018-05058-y> - WGS - 5/11 cases showed translocations of MYC

- We thank the reviewer for their comment and feedback. We agree with the reviewer that we were limited in the detection of *MYC* translocations as FISH was the method to detect in the primary cohort. For *MYC* CNAs, we were able to detect these aberrations confidently as we performed WES in the majority of samples and were also able to call the CNAs in the target panels. After performing the binary matrix factorization and consensus clustering on the 214 samples, *MYC* translocation was found to be enriched in the HL3 cluster ($p = 0.0037$, adjusted $p = 0.060$). As requested, we have provided a table of the significantly enriched genetic alterations in each cluster in the new added Supplementary table 1B. We agree and acknowledge that although this feature was statistically significant based on our preassigned cutoffs for p and q values outlined in the methods section, the small number prevents a definitive conclusion. To address this limitation, we added the following paragraph in the discussion section highlighting the limitations of the study (page 15). “Another limitation is that we depended on FISH studies in assessing *MYC* translocations in the primary cohort. FISH studies are less sensitive in detecting *MYC* translocations compared to novel targeted sequencing panels. Indeed, the validation cohorts, which used a targeted NGS panel in detecting *MYC* alterations, had more events compared to ours, suggesting that further studies that detect *MYC* with next generation sequencing panels in SMM are needed to delineate the characteristics of tumors harboring this important feature. However, *MYC* alterations was a prognostic factor for progression in the primary and second validation cohorts, as well as the combined cohort.”

4. I agree with reviewer 1 that biologically, findings are compatible with previous work identifying high risk in myeloma, rather than identifying novel prognostic biology, and generally support others' findings that SMM is really genetically very similar to symptomatic MM.

However, I agree with authors that a DNA-only based classifier is helpful as this is more accessible to most clinicians than RNASeq. Nonetheless it could still be clearer for the non-expert reader, where BMF approach is described early in results, that the transcriptomic data does not contribute to the clustering of the primary cohort, particularly

as it is referred to as 'multi-omics' subtyping: "We instead apply consensus BMF for this multi-omics subtyping" - this is a bit misleading.

- We thank the reviewer for their note and we updated the text and remove multiomics subtyping to avoid confusion. We also previously provided Figure 1A as a scheme for our approach.

5. As above, I find the concept that clustering was done using DNA panel-detected genomic events helpful, as this is more accessible to most clinicians than RNASeq, in contrast to previously-published gene-expression-based clustering, as authors point out. Fig 1C is helpful to this end, as is the determination of the minimum classifiers required. Supp Fig 5 not referenced in the text?

- We are glad we addressed the reviewer's comments and we also now updated the text to reference Supp Fig 5. We changed the order of the supplementary figures, so it is now cited in page 11, as Supplementary Figure 4.

6. I am happy with this answer and agree the t(14;16) cohort is too small for conclusive association analysis.

7. Fine

Extra small comments

1. Where clusters are described on page 7, the sentence 'We named this cluster Translocation-like1 (TL1).' Is used in reference to cluster 3 AND cluster 5 - typo.

- We fixed this typo and updated the text accordingly.

2. Gene names Fig 1B are poorly legible

- We have uploaded high resolution files for all the main figures, and they will be available with the revised manuscript.

3. There seems to be an error where Fig. 2 has been updated but the text referring to it has not been? 2A-E references in the main text do not match and the quoted p values do not match the figure? Also confusion between 'HP' and 'HL' label in legend. 2F/G called 'FMD' and 'CND' in legend but 'TL1' and 'TL2' on the figure is confusing. There are 2 'G's.

- We thank the reviewer for this note. We now corrected all the mentioned typos.

4. Even enlarging 500% on my screen I cannot read the gene names in 2F and G!

- We have uploaded high resolution file for all the main figures, and they will be available with the revised manuscript.

5. Newly added sentence about supp table 4 (mis-referenced as 3) - this table suddenly introduces extra validation cohorts before the appropriate text section explaining them, which is confusing

- We thank the reviewer for noting this issue. We fixed the order and numbering of the supplementary tables and figures. We removed this referenced table from the above-mentioned paragraph and it is now referenced as supplementary table 4 when we introduced the validation cohorts in page 12.

6. Several of the supp figures are not referenced in the text / not ordered as per reference in text.

- We changed the order of the supplementary figures and tables, and they are all now referenced in the updated manuscript and methods section.

Reviewer #5

assessed responses to Reviewer's #2 previous requests (Remarks to the Author):

The concerns were addressed satisfactorily. However, it is unclear whether the P values provided for the significance of the APOBEC activity signature were adjusted for multiple testing. If not, the authors should include the adjusted data.

- We are glad we satisfactorily addressed the reviewer's comments and thank them for feedback. We provide both the naive and adjusted p values to the updated text in page 9.